# Cryo-EM led analysis of open and closed conformations of Chagas vaccine candidate TcPOP

Sagar Batra[1,2,11], Francisco Olmo [3,11], Timothy J. Ragan [4], Merve Kaplan[5,6], Valeria Calvaresi [6,7], Asger Meldgaard Frank[8], Claudia Lancey [4], Mahya Assadipapari [9], Cuifeng Ying[9], Weston B. Struwe [6,7], Emma L. Hesketh[4], John M. Kelly [10], Lea Barfod[8] & Ivan Campeotto [1] ✉

Chagas disease, caused by the protozoan parasite *Trypanosoma cruzi*, remains a significant global public health concern. Despite its profound health impact in both endemic and non-endemic areas, no vaccine is available, and the existing therapies are outdated, producing severe side effects. The 80 kDa prolyl oligopeptidase of *Trypanosoma cruzi* (TcPOP) has been identified as a leading candidate for Chagas vaccine development. Here we report the three-dimensional structure of TcPOP in open and closed conformation, at a global resolution of 3.8 and 3.6 Å, respectively, determined using single-particle cryo-electron microscopy. Multiple conformations were observed and further characterized using plasmonic optical tweezers and hydrogen-deuterium exchange mass spectrometry. To assess the immunogenic potential of TcPOP, we immunized female mice and evaluated both polyclonal and monoclonal responses against the TcPOP antigen and its homologues. The anti-TcPOP polyclonal response demonstrates invasion blocking properties via parasite lysis. Polyclonal sera were cross-reactive with closely-related POPs but not with human homologues. Collectively, our findings provide structural and functional insights necessary to understand the immunogenicity of TcPOP for future Chagas vaccine development.

Chagas disease is a chronic life-threatening parasitic infection caused by *Trypanosoma cruzi* and represents a significant health burden in 21 countries of Latin America[1]. Chagas continues to expand beyond endemic zones because of human migration and global warming[2], with 6 million infected people worldwide, leading to ~12,000 deaths per year (PANHO[3]). The major transmission route is via the bite of insects belonging to the Triatomine species in endemic regions, although other routes include congenital

[1]School of Biosciences, Division of Microbiology, Brewing and Biotechnology, University of Nottingham, Sutton Bonington Campus, Loughborough, UK. [2]Interdisciplinary Biomedical Research Center, School of Science and Technology, Nottingham Trent University, Nottingham, UK. [3]Department of Parasitology, Faculty of Sciences, University of Granada, Granada, Spain. [4]Leicester Institute of Structural and Chemical Biology, University of Leicester, Leicester, UK. [5]Physical and Theoretical Chemistry Laboratory, Department of Chemistry, University of Oxford, Oxford, UK. [6]Kavli Institute for Nanoscience Discovery, Dorothy Crowfoot Hodgkin Building, University of Oxford, Oxford, UK. [7]Department of Biochemistry, University of Oxford, Oxford, UK. [8]Department of Immunology and Microbiology, Centre for Medical Parasitology, Faculty of Health and Medical Sciences, University of Copenhagen, Copenhagen, Denmark. [9]Advanced Optics & Photonics Laboratory, Department of Engineering, School of Science & Technology, Nottingham Trent University, Nottingham, UK. [10]Department of Infection Biology, Faculty of Infectious and Tropical Diseases, London School of Hygiene and Tropical Medicine, London, UK. [11]These authors contributed equally: Sagar Batra, Francisco Olmo. ✉e-mail: ivan.campeotto@nottingham.ac.uk

transmission, organ transplants, blood transfusion, or oral transmission[4].

Chagas disease becomes symptomatic during the chronic stage when muscle cells in the heart and gut are compromised by both infection and immune responses against the parasite; this results in cumulative damage that leads to severe cardiac complications and heart failure in ~30% of patients, with an estimated global economic hardship of $7.19 billion US per year[5]. Chagas disease is, therefore, one of the major global neglected

tropical diseases (NTDs). There is no vaccine, and current therapy relies on the usage of the drugs nifurtimox and benznidazole[6], which can cause severe side effects including sterility, blindness, and deleterious effects in adrenal, colon, oesophageal, and mammary tissue[7]. Additionally, therapies are more effective during the acute phase of the disease, and less so during chronic phase. Infection with *T. cruzi* is generally considered to be lifelong, therefore representing a time bomb on health systems around the world.

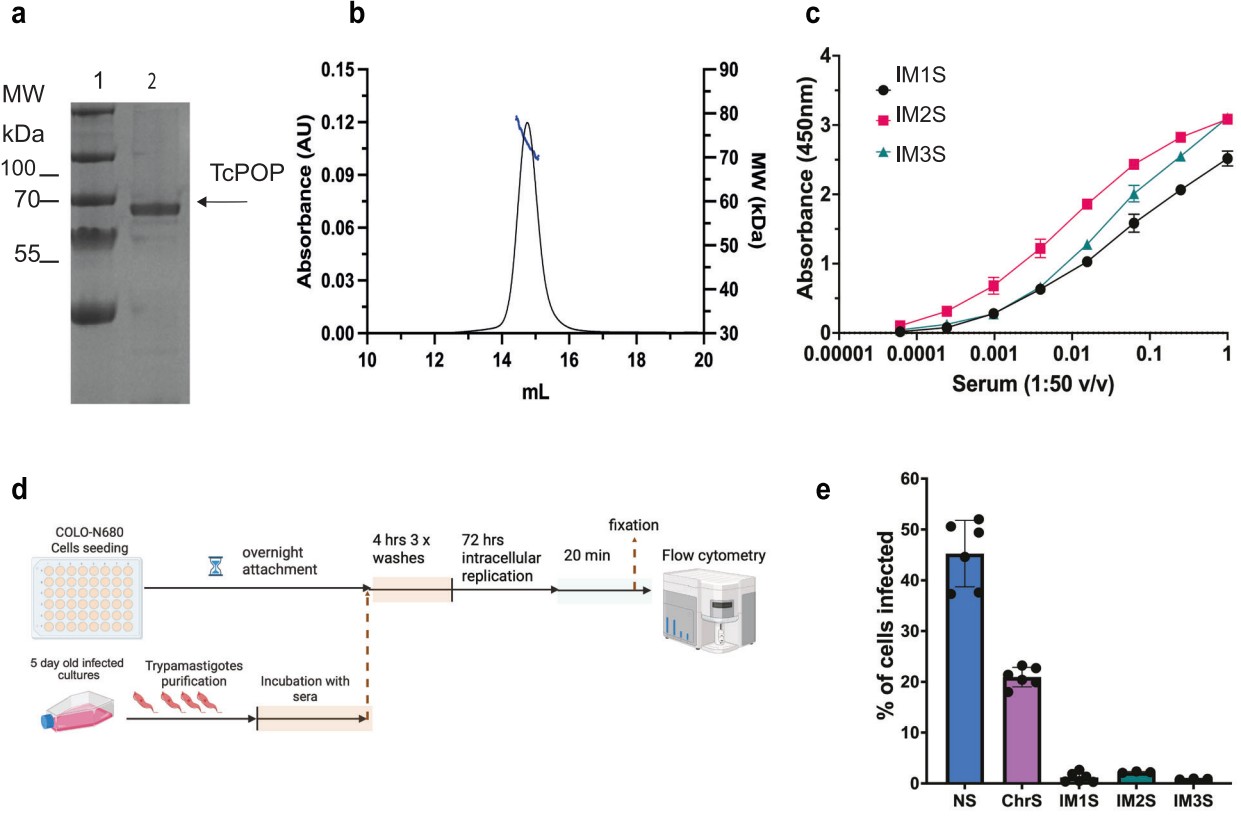

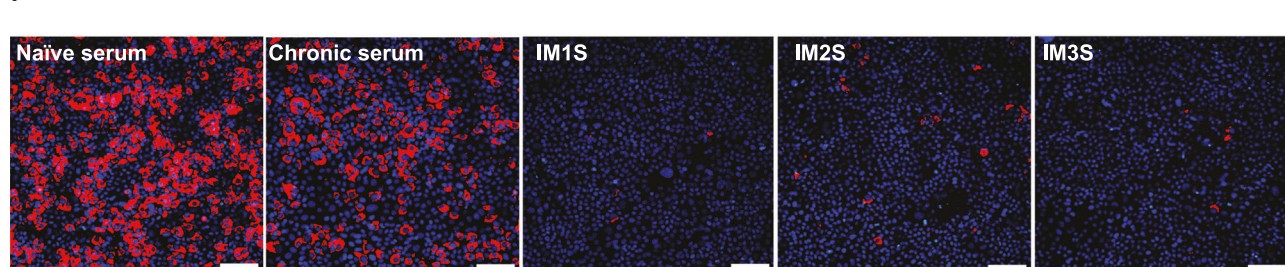

**Fig. 1 | Expression and purification of recombinant prolyl oligopeptidase from *Trypanosoma cruzi* (TcPOP) and trypomastigote-inhibitory activity of murine anti-sera. a** Analysis of TcPOP samples by SDS-PAGE: 1 = MW ladder (kDa), 2 = TcPOP. **b**, SEC-MALS UV chromatograms of TcPOP using a S200 10/300 column. **c** ELISA testing of anti-TcPOP polyclonal sera from three immunised mice (IM1, IM2, IM3). Polyclonal serum from an initial dilution (1:50 v/v) was further diluted (10 to $10^5$ times) and tested against recombinant TcPOP by measuring absorbance at 450 nm. Experiments were performed in triplicates. The data with error bars represents standard error of the mean (SEM). **d** Experimental design of the in vitro neutralisation assay used to evaluate infection blocking activity of anti-TcPOP polyclonal sera created with BioRender (https://BioRender.com/izeudu7 academic licence). **e** Blocking effect on trypomastigote infectivity. Trypomastigotes were

exposed to immunised sera for 4 h, followed by infection of COLO-N680 cells. The percentage of infected cells was determined 72 h post-infection. Sera from immunised mice (IM1S, IM2S, and IM3S) were compared with naïve serum (NS) and serum from a chronically infected mouse (>100 days) (ChrS). Data are presented as mean ± SEM from three independent biological replicates with two technical replicates per group, and statistical significance was assessed using Kruskal–Wallis test followed by Dunn's multiple comparisons test ($P < 0.05$). **f** Representative images showing infection density. *T. cruzi* intracellular amastigotes (red) captured using live fluorescent microscopy, after DNA-Hoechst staining (blue), and prior to trypsinization. This experiment was performed three times with similar results. Scale bars = 100 μm.

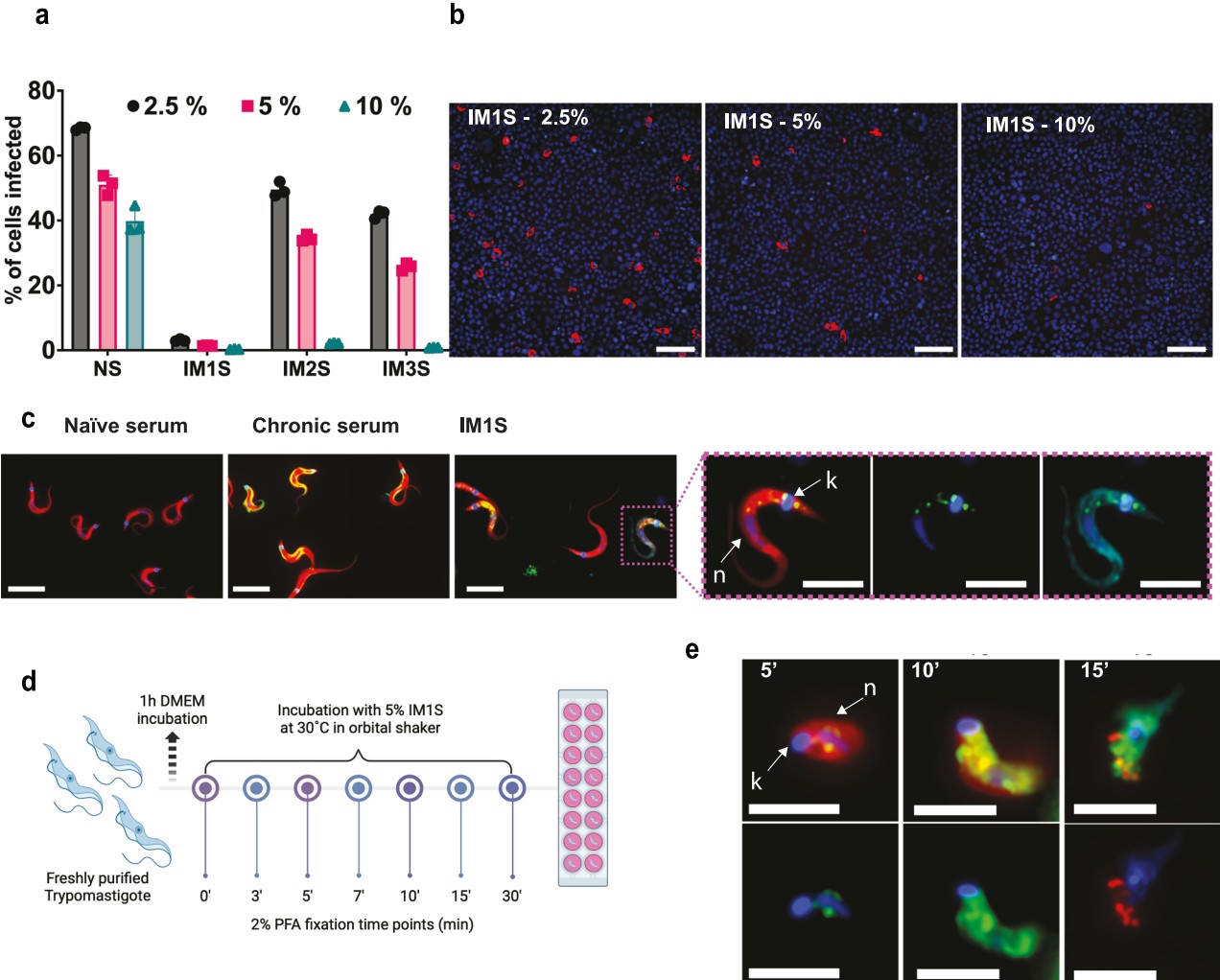

**Fig. 2 | Blocking activity of polyclonal immune serum IM1S significantly reduces in vitro infection by exerting a lytic effect on trypomastigotes.**
**a** Trypomastigote-blocking effects on COLO-N680 cell infection assessed using sera from three mice immunised with TcPOP. Antisera were added at three concentrations and compared with naïve serum. Infection reduction is presented as the percentage of infected cells 72 h post-infection with trypomastigotes exposed to antisera for 4 h (Fig. 1d). Data are presented as mean ± SD from three independent biological replicates with two technical replicates per group, and statistical significance was assessed using two-ways ANOVA with Dunnett's multiple comparisons test for the percentage of serum effect (*) and Turkey's multiple comparisons test for the effect due to the percentage of serum within the same polyclonal serum.
**b** Representative images, after performing the experiment three times with similar results, showing infection density (*T. cruzi* intracellular amastigotes, red) under live fluorescent microscopy after host cell DNA-Hoechst staining (blue), and before trypsinisation. Images depict infections following incubation with various concentrations of polyclonal antisera IM1S and naïve serum. Scale bars = 100 μm.
**c** Representative images display IgG binding in polyclonal sera, visualised through immunostaining using anti-mouse IgG-AF488 (green) in fixed and saponin-permeabilized trypomastigotes (red), with DNA stained using DAPI (blue). Nuclear

(N) and kinetoplast (k) DNA are indicated by arrows. Naïve serum displayed no binding (negative control). Serum from a chronically infected (>100 days) mouse exhibited a dispersed binding pattern (positive control). Antiserum IM1S demonstrated high affinity for TcPOP in the secretory pathway, consistent the reported location. The zoomed image of a single trypomastigote shows a binding pattern within the parasite, corresponding to characteristic vesicles of the secretory pathway. The trypomastigote surface is highlighted by Wheat germ agglutinin staining (turquoise). Scale bars = 5 μm. (see Supplementary-movie 2.mov). **d** Schematic representation of the in vitro polyclonal antibody live-binding assay designed to capture the intermediate lysis effect induced by antiserum IM1S. Image produced with Biorender (https://BioRender.com/9tt05k9, academic licence)
**e** Representative images from modified live-binding assays show experimental conditions where incubation temperature was reduced to 30 °C and serum concentration was 5% antiserum + 5% FBS At 5 min post-incubation, antibodies are detected binding to TcPOP in endocytic/exocytic vesicles near the flagellar pocket of the parasite, and trypomastigotes appear swollen. By 10 min, trypomastigotes exhibit internal leakage in multiple vesicles, followed by complete lysis by 15 min. Scale bars = 5 μm. All the microscopy experiments shown in this figure were performed independently three times obtaining similar results.

To complicate the epidemiological scenario, due to the evolutionary plasticity of *T. cruzi*, there are six discrete typing units (DTUs)[8] found to infect humans. An additional DTU has been identified in bats (Tcbat), although transmission to humans has not yet been reported[9]. DTUs differ in geographical and ethnic distribution, clinical manifestations, and reservoir hosts, which include more than 150 species of mammals[10].

Diagnostic tools are often strain-specific and unable to detect congenital transmission in newborn babies for the first 6 months, due

to passive immunity from the mother. Neonatal infections can cause life-threatening complications later in life (PANHO[3]). Amidst these challenges, a member of the endopeptidase enzyme family, *T. cruzi* 80 kDa prolyl-oligopeptidase (TcPOP, enzyme ID EC 3.4.21.26), has been identified as a potential vaccine target for Chagas disease[11]. TcPOP is expressed in both the extracellular blood trypomastigote and the replicative intracellular amastigote[12]. It degrades collagen and fibronectin extracellular matrix components on host cells, facilitating parasite invasion. Its secretion into the blood during invasion, and

its expression throughout the mammalian stages of the life cycle, and the more than 98–99% sequence conservation across DTUs[11] makes it an ideal candidate for vaccine development. This has been further supported in a murine model, as polyclonal antibodies against recombinant TcPOP, produced in *E. coli*, protected the mice from a lethal dose of the parasite[11]. *T. cruzi* triggers a complex immune response in humans, involving both innate and adaptive systems. Initially, the innate response is activated, including cytokine production by phagocytes and natural killer cells, which produce interferon-gamma[13]. The adaptive response involves T and B lymphocytes, with T cells orchestrating cellular immunity and B cells producing antibodies, although these are often ineffective due to the parasite's evasion strategies[14]. *T. cruzi* employs mechanisms including hijacking of the TGF-β signalling pathway and modulating immune responses to evade detection and establish a chronic infection. Achieving a potent and protective humoral response would be an ideal approach to block transmission in humans[15].

No structure of parasite prolyl oligopeptidases (POPs) exists to date, despite POPs being reported in *Leishmania infantum* (LiPOP,[16]), *Trypanosoma brucei (*TbPOP)[17], and *Schistosoma mansoni* (SmPOP)[18]. The POP family is widely distributed across Eukaryotes and Prokayotes[19]. The closest protein homologues to TcPOP for which experimental structures are available are from *Haliotis discus hannai* (PDB code 6JCI), porcine muscle POP (PDB code 1QFM), *Pyrococcus furiosus* (PDB code 5T88), and human POP (PDB 3DDU). Comparative homology modelling studies of TcPOP have been based on porcine POP[20] and predict a cylindrical-shaped structure, consisting of a peptidase domain and a seven-bladed β-propeller domain with the substrate binding site and the canonical Asp, Ser, His catalytic triad located in the middle of the two domains[21].

The TcPOP sequence is highly similar to LiPOP (63%), TbPOP (73%), and HuPOP (43%). We therefore also expressed these other POPs and immunised mice against TcPOP to investigate cross-reactivity related to epitope conservation across species. Polyclonal antibodies (pAbs) were isolated and characterised from mice and showed striking parasite-neutralising properties in cell-invasion assays. Additionally, one monoclonal antibody was also identified from the mice exhibiting a neutralising response. The structure of TcPOP was solved by cryo-EM in closed and open conformations, revealing unexpected details on its dynamicity, corroborated by a plethora of in-solution biophysical and structural techniques, which offer insights into the dynamicity of TcPOP, paving the way for new therapeutic interventions against Chagas disease.

## Results

### Expression and purification of TcPOP and TcPOP homologues

Prolyl oligopeptidases (POPs) belong to the large hydrolase family, which is highly conserved in eukaryotes. To explore the evolutionary and functional similarities, we compared *T. cruzi POP* (TcPOP) to two close parasite homologues, namely those in *T. brucei* (TbPOP), *L. infantum* (LiPOP), and the *Homo sapiens* homologue (HuPOP). Phylogenetic analysis and sequence conservation mapping on TcPOP highlighted that conserved and variable regions are located in both the catalytic and non-catalytic domains (Fig. S1a). Notably, the AlphaFold3 modelling of collagen binding to TcPOP corroborated previous docking studies with the substrate-binding site at the interface of the two domains (Supplementary-movie 1.mp4). This is a region of high variability, as indicated by the phylogenetic analysis (Fig. S1b).[21] TcPOP was expressed in *E. coli* using bacterial fermentation, overcoming the challenge of extremely low expression yields in batch culture (~0.05 mg/L) (Fig. 1a). In contrast, expression of the other homologue proteins led to higher protein yields (0.8-2.0 mg/mL) (Fig. S2a). Following purification by size-exclusion chromatography (SEC) (Fig. S2b), all POPs exhibited enzymatic activity (Fig.

S3a–c), thermostability, as determined by differential scanning fluorimetry (DSF) (Fig. S3d), and comparable end-point kinetics consistent with previously reported values for this enzyme family[22]. SEC-MALS analysis confirmed that all proteins were monomeric and monodisperse in solution (Fig. 1b) and predicted the expected mass within the experimental error of the technique (TcPOP: 73.4 (±2.1%) kDa, TbPOP: 73.7 (±1.6%) kDa, LiPOP: 75.7 (±3.5%) kDa, HuPOP: 85.3 (±1.71%) kDa) (Fig. S4a). AlphaFold3 modelling of TcPOP predicted a structure comprising an α/β-hydrolase domain housing the catalytic triad Ser548-Asp631-His667, and a seven-bladed beta-propeller non-catalytic domain (Fig. S5), consistent with the previously determined structure of porcine POP (PDB code 1QFM)[21].

### Anti-TcPOP polyclonal response is cross-reactive against other trypanosmatid parasites

To assess the cross-reactivity of antibodies raised against TcPOP, mice were immunised with the recombinant enzyme. Homologous proteins with high amino acid identity from *L. infantum* (LiPOP, 63%), *T. brucei* (TbPOP, 73%), and *H. sapiens* (HuPOP, 43%) were expressed and purified to evaluate potential cross-reactivity, driven by the likelihood of shared epitopes (Fig. S1a). Three mice were immunised with TcPOP, and polyclonal response analysed by ELISA (Fig. 1c). The corresponding TcPOP-specific sera were tested against all four POPs. Cross-reactivity was detected to TcPOP, TbPOP, and LiPOP, but not for HuPOP at serially diluted serum concentrations. These findings suggest potential epitope conservation across parasite species, but not with HuPOP (Fig. S6a).

### Anti-TcPOP polyclonal sera block parasite invasion

To evaluate the neutralising properties of anti-TcPOP polyclonal sera, we adapted our trypomastigote drug screening procedure[23] to a cell-based invasion assay (Fig. 1d). Polyclonal antisera from all three mice demonstrated significant neutralising activity, greatly reducing trypomastigote invasion of COLO-N680 cells compared to the naïve serum and serum derived from mice chronically infected with *T. cruzi*. (Fig. S7a). Overall, we found that all three independently derived sera achieved a >95% neutralising effect, an impact that approached 99% with the serum from mouse 1 (IM1S (Fig. 1e, f), as measured by flow cytometry 72 h post infection (Supplementary Fig. S8a, b). The potency of IM1S was maintained even when reduced to 2.5% (Fig. 2a, b and Fig. S7b).

Immunofluorescence analysis of freshly extracted and fixed parasites revealed that the anti-TcPOP antisera bound in the endocytic/exocytic pathway of trypomastigotes (Fig. 2c, Supplementary Fig. S9b and Supplementary Movie 2.mov). This pattern differed from antiserum from chronically infected mice, which showed the expected dispersed binding on the parasite surface, associated with a non-protective response (Fig. 2c). To understand the dynamics of IMIS binding, we performed a real-time assay, which revealed that antibody attachment led to parasite lysis within minutes. To dissect this effect, we monitored the process by reducing the amount of serum and decreasing the temperature from 37 °C to 30 °C. (Fig. 2d and Supplementary Fig. S9a). Even under these constrained conditions, parasites became swollen after 5 min exposure, followed by intracellular leakage and parasite lysis within 15 min. (Fig. 2e and Supplementary Fig. S9c). To confirm this lytic effect, we incorporated a competition assay, including foetal bovine serum (FBS) at the same concentration as the IM1S. At the early timepoints, parasite integrity was preserved, and the majority of binding was restricted to the parasite surface. (Supplementary Fig. S9d).

Additionally, we investigated binding to amastigotes, the intracellular form of the parasite. This life-cycle stage is also infectious, for example, when are released as host cells rupture following trypomastigote egress. The amastigote binding profile also highlighted the

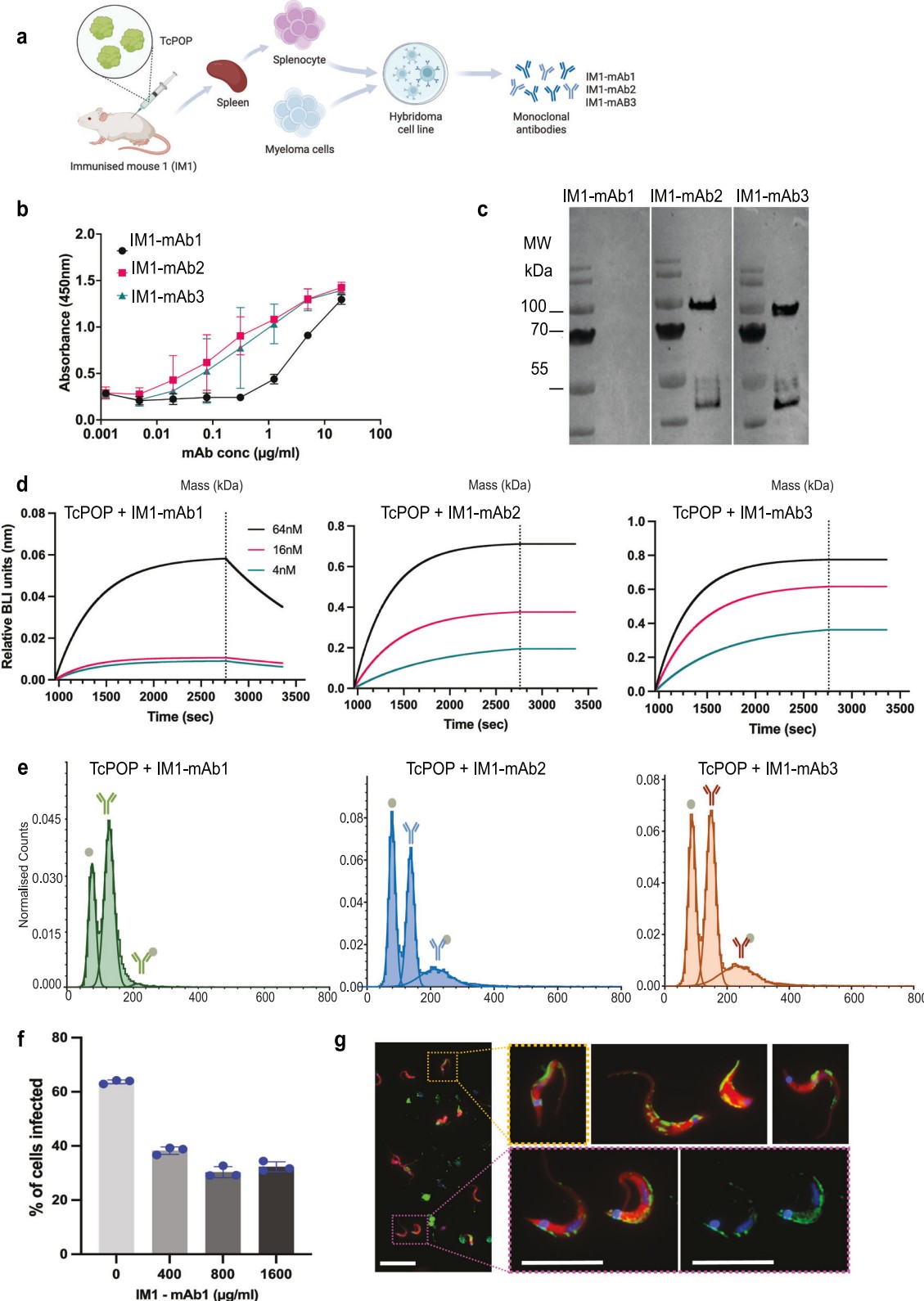

endocytic/exocytic pathway (Supplementary Fig. S9b). However, in live-binding assays, we also observed that surface attachment of anti-TcPOP, located at the apical end of the amastigote (Supplementary Fig. S9c, d). Finally, we determined binding to epimastigotes, the replicative insect form of the parasite The main location was at the anterior end, adjacent to the cytostome-cytopharynx complex, an additional secretory portal found in this form of the parasite.

(Supplementary Fig. S9c, d). Overall, the binding studies reveal that TcPOP antisera is associated predominantly with the secretory pathway in all forms of the parasite.

## Anti-TcPOP monoclonal response
Three anti-TcPOP monoclonal antibodies (mAbs) were generated from the spleen of the mouse with the highest polyclonal serum response

**Fig. 3 | Anti-TcPOP monoclonal antibody production and characterisation.**
**a** Schematic for neutralising monoclonal antibody production. Image produced with Biorender (https://BioRender.com/cc1ow7b, academic license). **b** ELISA testing of monoclonal antibodies from mouse 1 (IM1-mAb1, IM1-mAb2 abd IM1-mAb3) against recombinant TcPOP. mAbs concentrations were tested in the 0.001–1 μm/mL range by measuring the absorbance at 450 nm. Experiments were performed in triplicate. The data with error bars represents standard error of the mean (SEM). **c** Western blot analysis with the three mAbs from mouse 1, using secondary rabbit anti-mouse HRP-conjugate, verifies in vitro specificity against purified antigens. **d** Binding affinity of IM1-mAb1, IM1-mAb2 and IM1-mAb3 to TcPOP by biolayer interferometry (BLI). The association and dissociation of the response curves are shown. The black, red and green lines represent the fitted curves based on the experimental data. Equilibrium dissociation constants ($K_d$) are shown and colour-coded **e** Mass photometry analysis of complex formation between TcPOP andIM1-mAb1 (green), IM1-mAb2 (blue), or IM1-mAb3 (orange). **f** Assessment of the

blocking effect on trypomastigote infections. Trypomastigotes were exposed to the indicated concentration of M1-mAb1 in DMEM for 4 h, followed by an infection challenge in vitro using COLO-N680 cells. The percentage of infected cells was determined 72 h post-infection. Data are presented as mean ± SD from three independent biological replicates with two technical replicates per group, and statistical significance was assessed using ordinary one-way ANOVA test followed by Dunnett's comparisons test ($P < 0.05$). **g** Representative images from live-binding assays where freshly isolated trypomastigotes were incubated with 1000 μg/ml of IM1-mAb1 in DMEM prior to fixation. Then parasites were blocked and stained with anti-mouse IgG-AF448 to determine the binding site for the antibodies. Superficial binding across the flagellum line was the most common pattern observed (see Supplementary Movie 3.mov). All the microscopy experiments shown in this figure were performed independently three times. Scale bars from left to right = 20, 10, and 10 μm, respectively.

(Fig. 3a) and screened by ELISA for specificity against TcPOP (Fig. 3b) and homologue POPs (Fig. S6b). TcPOP antibody titres were positive at 1 μg/ml for mAb1 and at 0.1 μg/ml for mAbs 2 and 3 (Fig. 3b), while no binding was observed against LiPOP, TbPOP, and HuPOP. HRP-conjugation of anti-TcPOP mAbs revealed the presence of both linear and conformational epitopes, as confirmed by Western blot analysis (Fig. 3c).

The BLI-based binding profile for immobilised anti-TcPOP mAbs against TcPOP showed higher binding affinity to TcPOP for anti-TcPOP mAb2 ($K_D < 1$ pM, $K_a$ (1/Ms) $2.3 \times 10^5$, $K_{diss}$ (1/s) $< 10^{-7}$) and anti-TcPOP mAb3 ($K_D < 1$ pM, $K_a$ (1/Ms) $3.1 \times 10^5$, $K_{diss}$ (1/s) $< 10^{-7}$). Instead, anti-TcPOP mAb1 exhibited comparatively weak binding ($K_D < 3.2$ nM, $K_a$ (1/Ms) $7.1 \times 10^4$, $K_{diss}$ (1/s) $6.2 \times 10^{-4}$) (Fig. 3d), as well as association and dissociation parameters. The best two mAbs, mAb2 and mAb3, were used to perform a sequential affinity binding with association and dissociation lengths of 1800 s and 600 s, respectively. Both mAbs showed detectable association and dissociation ($K_D < 1$ pM, $K_a$ (1/Ms) $> 10^7$, $K_{diss}$ (1/s) $< 10^{-7}$), indicating competition for different epitopes (Fig. S11c).

Mass photometry measurements of TcPOP and the three individual anti-TcPOP mAbs were 85 kDa and 151 kDa, respectively, with only a single peak present in each MP spectrum (Fig. S11b). Interaction studies revealed similar binding between TcPOP and both antibodies. A peak corresponding to 1:1 (mAb to TcPOP) was observed at 233 kDa with a relative abundance of 1-2% (TcPOP-mAb1), 17% (TcPOP-mAb2), and 19% (TcPOP-mAb3). No evidence of higher-order binding was found (Fig. 3e).

### Anti-TcPOP monoclonal antibody neutralises parasite invasion
To further investigate the neutralising activity of the mAbs, we performed a cell-based infection assay, incubating the parasites in 1000 μg/ml of each purified antibody in DMEM. Notably, mAb1 caused a significant reduction in parasite invasion, approaching 50% (Fig. 3f and Supplementary Fig. S10a). Additionally, we found that this neutralising effect was maintained regardless of the concentration assayed (Fig. 3f), and that instead of inducing a lytic effect, the effect was mediated by steric hinderance resulting from the binding to the surface of the trypomastigotes (Fig. 3g). The colocation assay showed a specific binding in a line along the flagellum (Supplementary Fig. S10b, c, Supplementary Movie 3.mov). Additionally, we observed that in amastigotes, the binding also associated with the short flagellum, that in these rounded forms of the parasite is largely confined to the flagellar pocket (Supplementary Fig. S10b, c).

### Determination of cryo-EM structure of TcPOP in multiple conformations
**TcPOP structure determination using single particle cryo-EM.** We analysed the structure of TcPOP using single particle analysis (SPA) cryo-EM. 2D classification highlighted equal distribution of the

particles in the micrographs (Fig. S12). The initial data collection, comprised of 20,080 micrographs, revealed severe preferential orientation[24], resulting in poor map reconstructions (Fig. S13). Several approaches were attempted in cryo-EM grid preparation, including changes in detergent, support films, and glow discharge parameters. However, none led to significant improvement in the orientation distribution. We therefore tilted the grids to 30° and 35° to increase the number of views visualised in the electron micrographs (Fig. S13). This significantly improved the number of views, and the resulting map was drastically improved with continuous density and minimal anisotropic features (Fig. S13). 3D classification revealed two distinct conformations of TcPOP i.e., open and closed (Fig. S13), indicating intrinsic conformational heterogeneity of the dataset. Closed and open conformation models obtained from AlphaFold and homology modelling (using PDB 3IUJ), respectively, were then docked into their respective cryo-EM maps and refined (Fig. 4a, b and Supplementary Table Data 1). Local resolution was also estimated using half-maps in PHENIX (Fig. 4a, b).

Our cryo-EM data confirm that TcPOP comprises two distinct domains: a catalytic α/β-hydrolase domain, which houses the active site featuring the catalytic triad, and a β-propeller domain, which likely regulates substrate access. The spatial arrangement of these domains reveals a cylindrical structure, with the β-propeller domain capping the catalytic α/β-hydrolase domain, as previously postulated[25]. This configuration suggests a gating mechanism controlling substrate entry to the active site, which may play a critical role in modulating enzymatic activity. These findings align with the proposed role of TcPOP in hydrolysing large substrates and facilitating parasite invasion into host cells. Transition between open and closed conformation is reported as a Movie, morphing cryo-EM maps from closed to open conformation (Supplementary Movie 1).

### SAXS and mass photometry analysis
SEC-SAXS profiles on purified TcPOP indicated no aggregation and globular conformation, leading to an estimated molecular weight of 73.4 kDa[26], based on Kratky plot and Guinier analysis. Open and closed conformations were also deconvoluted, and the corresponding cryo-EM models were docked into the molecular envelopes (Fig. 4c). Static SAXS data were also collected in a range of concentrations, including zero-concentration, to determine the optimal level for SAXS data analysis and for data processing (Fig. S15). The discrepancy between the Rg values from Guinier (38 Å) and P(r) analysis (71 Å) in the static SAXS measurements is due to the presence of larger multimers that persist even when extrapolating to zero concentration. As a result, extrapolating to zero concentration cannot fully recover the properties of a species that was never measured in isolation. Given these limitations, we produced the P(r) plots and we relied only on SEC-SAXS data, which directly allowed the isolation of the monomeric species and provides in this case a more reliable approach. Mass photometry

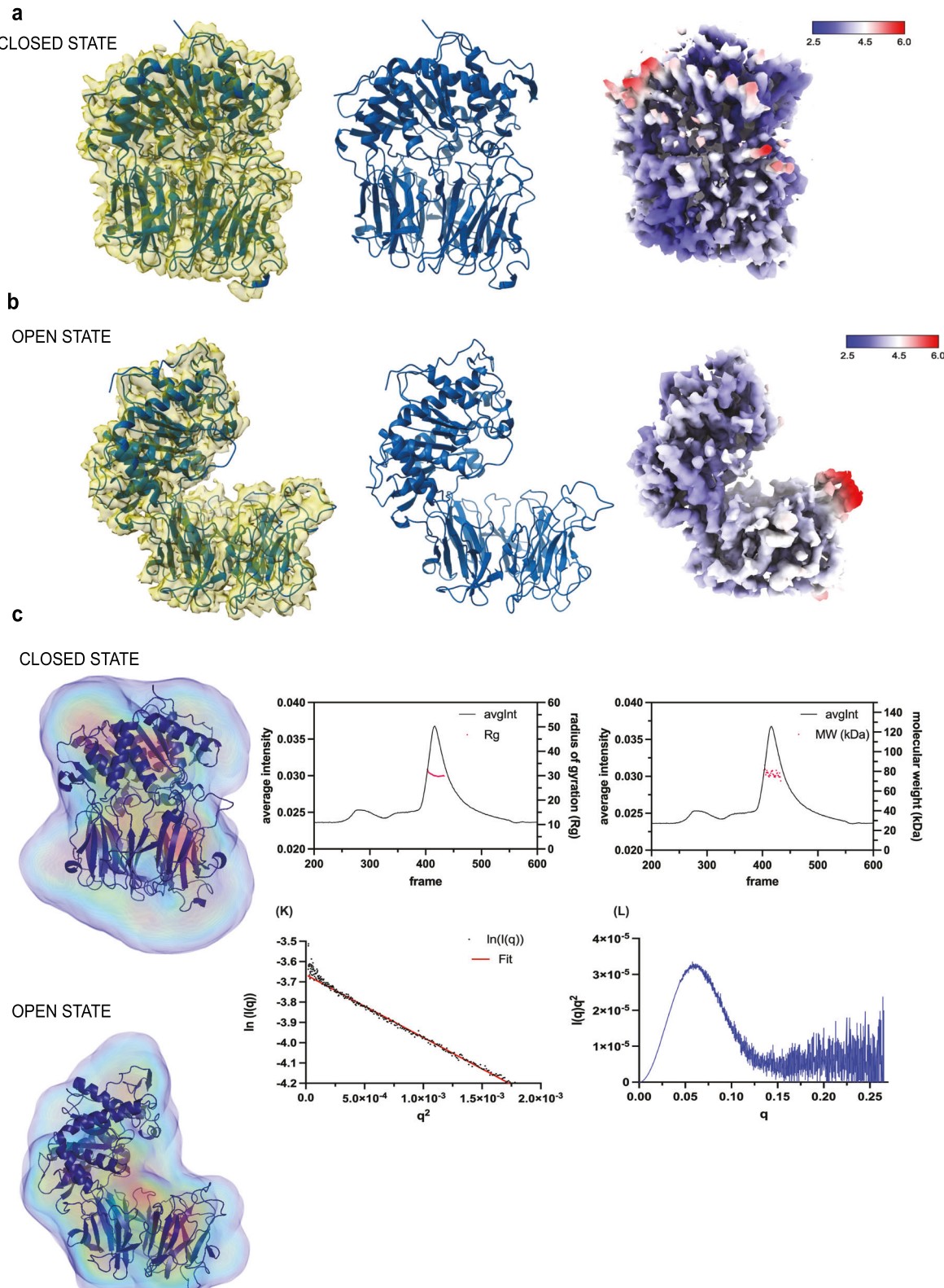

**Fig. 4 | Structure determination of TcPOP in the closed and open conformation.** Cryo-EM structures corresponding to (**a**) closed and (**b**) open states of TcPOP are shown for each lane as surface representation, cartoon representation, and local resolution estimates as surface, respectively, using ChimeraX. **c** SAXS density reconstructions of TcPOP in open and closed states, generated using DENSS, visualised using PyMOL. Colours represent the particles envelope. The elution profiles from the SEC-SAXS experiment representing the radius of gyration (Rg) and its corresponding estimated molecular weight, followed by Guinier analysis and Kratky plot.

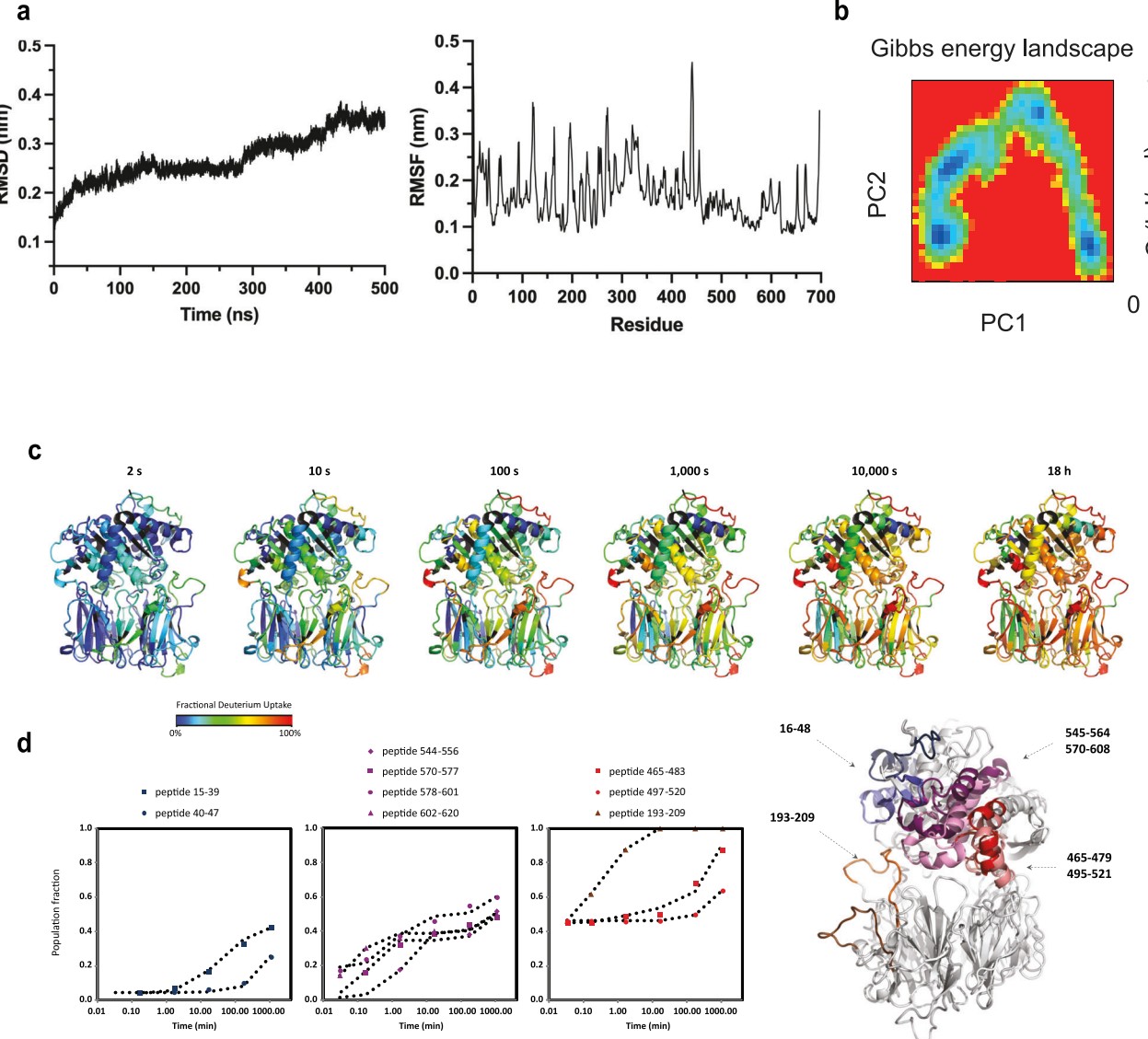

**Fig. 5 | Structural dynamics of TcPOP probed by MD and HDX-MS. a** RMSD and RMSF trajectories during 500 ns and Gibbs free energy landscape. **b** The local HDX of TcPOP relative to the MaxD control (% of fractional deuterium uptake) at the illustrated time points is colour-coded according to the rainbow palette and superimposed onto the structure of the close conformation of TcPOP, showing the local structural dynamics of the protein. **c** Plots illustrating the fraction of the high-mass population (representing the HDX-detected open conformation of TcPOP) over the time points studied for peptides manifesting EX1 or EXx kinetics. The peptides are grouped according to the kinetics of their opening events: blue: slow rate; purple: medium rate; red: fast rate; brown: super-fast rate. **d** The regions spanning residues determined as undergoing correlated exchange (EX1 or EXx kinetics) are coloured on the overlayed structures of the closed and open conformation of TcPOP, according to their rates of opening as determined by bimodal HDX fitting.

data also allowed identification of the best buffer conditions in terms of ionic strength and pH values (Fig. S11a). This information was exploited for cryo-EM grid preparation to increase sample homogeneity and also for buffer optimisation to elucidate TcPOP-mAbs interaction in BLI.

## MD simulation of TcPOP

To understand the intrinsic dynamicity of TcPOP, molecular dynamics simulations were performed using the AlphaFold model from Uniprot (Q71MD6) as the starting point for 500 ns simulations. These showed notable transitions in Root Mean Square Deviation (RMSD) around 290 nsec, while Root Mean Square Fluctuation (RMSF) (Fig. 5a) and Solvent Accessible Surface Area (SASA) and radius of gyration (Rg) (Fig. S16) underwent complementary contraction for the first 300 ns, followed

by significant expansion up to 500 ns, indicating substantial conformational changes. RMSF analysis identified residues 192–198 and 306–333 as highly flexible. The Gibbs free energy landscape analysis identified four significant minima basins, providing insights into local and global minima in a 2D and 3D projection of FEL, as described previously in ref. 27 (Fig. 5a).

## Plasmonic optical nanotweezers confirms TcPOP dynamicity in solution

Aperture-based plasmonic nano-tweezers revealed the conformational dynamics of single, unmodified TcPOP. TcPOP trapping events were detected by changes in transmission levels, filtered at a cut-off frequency of 1 kHz, whilst time traces of $\Delta T/T_O$ of TcPOP trapped in the absence of substrate AMC (blue and purple, Fig. S17), and with 100 μM

AMC substrate introduced to the trapping site. Addition of substrate at 100 μM concentration led to larger fluctuations in transmission, indicating TcPOP exhibiting greater dynamic fluctuations during enzymatic cycles than in its apo-state. However, in the absence of a substrate, distinct signal fluctuations were observed above the background of Brownian motions in the trapping well, suggesting free transition of multiple conformations of TcPOP in solution. This is further corroborated by distinct peaks in the probability density functions (PDFs) (Materials and Methods).

### Structural dynamic analysis of TcPOP by HDX-MS

To gain insights on the kinetics of transition between the closed and open conformation in TcPOP, we applied hydrogen-deuterium exchange (HDX) mass spectrometry (MS). HDX-MS enables monitoring the structural dynamics of proteins in solution, based on the rate at which individual backbone amide hydrogens exchange with deuterium atoms[28]. We followed the HDX of 197 peptides, covering 98.9% of the protein sequence (Fig. S18). The two domains of TcPOP proved to be similarly dynamic, both containing regions of rapid, medium, or fast HDX (Figs. 5b, S19 and S20). Remarkably, peptides of several helices and loops of the α/β-hydrolase domain and of a single loop of β-propeller domain (Figs. S18 and S21) exhibited bimodal isotopic envelopes with resolved high- and low-mass distributions, whose intensities interconvert over time: a clear sign of EX1 and EXx kinetics. This type of HDX behaviour is generally seen for groups of amides undergoing cooperative unfolding events (correlated exchange) and populating an open state for a sizeable time frame[28,29]. By pulse deuterium labelling, we established that those segments are not irreversibly unfolding over time (Fig. S21) thus TcPOP is in a dynamic equilibrium under our HDX conditions and every opening event (determined by the rate constant $k_{op}$) is reversible. The bimodal isotopic envelopes were fitted[30] to determine the number of amides involved in correlated exchange, the kinetic parameters $k_{op}$, and HDX half-life of the closed state, which report on the relative rates at which individual helices/loops of TcPOP transition to the open conformation (Figs. S22a–j and S23, and Table S1). We localised the correlated exchange in the regions spanning residues 16–48, 465–479, 495–521, 545–564, and 570–608 (α/β-hydrolase) and 193–209 (β-propeller domain) (Figs. 5d and S18). We observed that the loop 193–209 (β-propeller domain, at the interface with the α/β-hydrolase domain) and the helix 495–521 and adjacent loop 465–479 (α/β-hydrolase domain) manifest a starting open state of ~40% of the population, indicating that these specific segments adopt two initial non-interconverting distinct conformations, and subsequently very fast transition to the open state (average half-life <1000 min at 23 °C). All other regions (helices and loop spanning residues 16–48 and 545–608 of the α/β-hydrolase domain) have an initial open state that approximates to 0%, with segment 16–48 interconverting very slowly to the open state (average half-life >2000 min at 23 °C), and segment 545–608 interconverting at medium rate (average half-life between 1000 and 2000 min at 23 °C) (Figs. 5c and S23, Supplementary Table 1). Taken together, our HDX-MS data corroborate with the existence of the TcPOP open and closed conformations that could be resolved in cryo-EM, and with the presence of intermediate conformations that arise from the switching of selected helices and loops at different rates.

## Discussion

Structural data on *T. cruzi* antigens are desperately needed to guide vaccine development and design of diagnostic solutions. Such data are invaluable to initiate structure-based immunogen design strategies that focus the immune response towards key conserved epitopes[31] and to identify regions within the target protein suitable for drug design. TcPOP is one of the leading validated candidates for a Chagas disease vaccine, given its high level of sequence conservation across the six genotypes (or DTUs) of the *T. cruzi* species.

The three-dimensional structure of TcPOP remained elusive for almost two decades since a comparative homology model was proposed in 2005[21]. Several attempt to crystalise parasite POPs in our lab, including TcPOP, LiPOP, TbPOP, and HuPOP, did not succeed, despite the usage of starting protein concentrations up to 500 μM for crystallisation screening (equivalent to ~40 mg/mL). Closed conformations for TcPOP homologues have been previously reported using protein crystallography from porcine POP (PDB code 1QFM), *Pyrococcus furiosus* (PDB code 5T88), or HuPOP with an irreversible inhibitor (PDB code 3DDU), to lock the enzyme in the closed state. However, to our knowledge, neither fully open nor fully closed conformations of a parasite POP in a single solution (with or without the aid of additives) have been determined previously. High solubility was a characteristic of not only TcPOP, but also of other POPs reported in this manuscript, and this is likely to be linked to their function of being secreted in the blood to perform degradation of the extracellular host matrix. In fact, TcPOP possess 33 Lys residues, out of which 13 are located on the solvent accessible surface, a feature that likely prevents the crystal contacts required to promote crystallisation. Methods have been developed to reduce protein entropy by lysine methylation[32], but we opted not to use this strategy, as this could have compromised antigenicity by altering potential key residues involved in epitope-paratope interaction.

Immunising mice with recombinant TcPOP led to the production of polyclonal antibodies that showed cross-reactivity with *L. infantum* and *T. brucei* homologues, indicating conserved epitopes among these parasites. However, the lack of response to the human homologue led to the speculation that there might be a divergence in epitope conservation across species. This might be attributable to evolutionary pressure or possible immune suppression in the human host, although this remains to be established. Such observed cross-reactivity not only underscores the potential for developing cross-protective vaccines against related parasite species but also emphasises the precision in targeting pathogen-specific epitopes, while minimising cross-reactivity with human proteins for diagnostic applications. Here, we investigated the immune response in mice at both the polyclonal and monoclonal levels to TcPOP. Polyclonal antibodies from all three mice exhibited striking neutralisation effects, approaching 100%, and leading to rapid parasite lysis of the infectious trypomastigote stage. These findings consolidate TcPOP's central role as a leading candidate for a Chagas disease vaccine. This led us to attempt to isolate mAb(s) responsible of this lytic effect from the mouse where the highest neutralising polyclonal response was observed. We were able to produce and to characterise three mAbs and identify a single mAb (IM1-mAb1) responsible for ~50% of the neutralisation effects. Unexpectedly, this mAb, despite being neutralising, did not induce parasite lysis, suggesting the presence of an alternative neutralisation mechanism, potentially involving hindrance, as indicated by cell localisation experiments. Western blot experiments suggest that IM1-mAb1 recognises a conformational epitope, while ELISA, BLI, and immunofluorescence experiments (IFA) confirmed its specificity for TcPOP, both as recombinant purified protein or indirectly when expressed on whole parasites. Although direct detection was not verified in parasite lysates, IM1-mAb1 still retains diagnostic potential, although further assessment is beyond the scope of the present study. Recognition of the purified TcPOP does not preclude the possibility of cross-reactivity with other *T. cruzi* proteins, and this will require further confirmation.

Interestingly, the IM1-mAb1 was the weakest binder in vitro among the three isolated mAbs, exhibiting faster *kon* and *koff* rates, whilst the other mAbs exhibited much longer *koff* rates. This underscores the importance of testing all mAbs in a cell-invasion scenario that more closely reflects host-pathogen interactions, without excluding any based on in vitro studies alone, as interactions with the parasite may reveal neutralising potential. Additionally, further studies will be

required to identify those mAb(s) responsible of the striking neutralisation effects observed in the anti-TcPOP polyclonal serum (Fig. 1).

In our quest to determine the three-dimensional structure of TcPOP, the AlphaFold model of the closed conformation of TcPOP highlighted its dynamic nature, showing significant transitions in the Cα-backbone during the course of molecular dynamics (MD) simulations. Also, MD simulation suggested that such transitions occur between the two domains of TcPOP, exposing their catalytic site during enzyme activity. Importantly, the β-propeller domain contributed to a high degree of movement and instability, whereas, α/β hydrolase domain remained comparatively stable. Moreover, the Gibbs free energy landscape of TcPOP indicated scattered blue spots, representing four-to-six major local or global energy minima, and therefore, provided valuable insights into the presence of different metastable states (Fig. 5a). Coincidentally, the number of minima matched the number of classes in cryo-EM, possibly indicating that there might be a correlation between MD predictions and classes distribution, which will be further investigated for method development.

We, therefore, decided to analyse the structure using cryo-EM, despite the challenges to date of resolving sub-80 kDa molecules, due to low image contrast[33]. To ensure the quality of samples before data collection, we implemented a quality control pipeline, which included SEC-SAXS in solution studies followed by assessment of the best buffer conditions using mass photometry to screen for the best conditions for monodispersion, therefore increasing the chances of success. Our approach led to determination of the TcPOP structure to 3.6 Å and 3.8 Å, respectively, in closed and open conformation. This is one of the smallest cases, and at the highest resolution reported, in terms of resolving multiple enzyme conformations in a vitreous state by single-particle cryo-EM for this class of enzymes[34]. Two conformations of TcPOP could be easily identified in 3D classification, spanning from fully closed to fully open with an overall motion of ~22° between the two domains. The evident stability of the α/β hydrolase domain observed in both open and closed conformations, as well as in those from aligned cryo-EM maps, suggests possible regions that are exposed to the immune system when the enzyme is secreted in the blood, and therefore likely to include potential epitopes.

To confirm that these experimentally observed conformations exist in solution, optical tweezer studies were performed in the presence and absence of substrate-mimicking peptide. Recent advances in plasmonic nano-tweezers allowed us to sample conformational fluctuations of single proteins in solution[35]. Our data show that TcPOP clearly fluctuates between open and closed states independently from the presence of the substrate, confirming that the observed open and closed conformations occur in solution and are observed without the usage of any additive. However, the addition of the substrate-mimicking peptide affects the frequency at which the enzyme opens and closes.

By HDX-MS, we primarily observed that selected helices and loops of the α/β-hydrolase domain and a single loop of β-propeller domain (at the interface with the α/β-hydrolase domain) switch between the two conformations via cooperative unfolding/refolding events along the protein backbone, which are interpretable as long-lived perturbations of their secondary structure. This has been previously observed for other enzymes[29]. Such a kinetic regime allowed us to understand the structural elements of TcPOP switching between the closed and open conformation at different rates, which likely leads the protein to occupy a multitude of intermediate states between the fully closed and fully open conformation.

The intrinsic conformational heterogeneity in solution can pose great challenges to the structure determination by cryo-EM, as multiple conformations merge into a single conformation. This often leads to preferential orientation, eventually producing lower-resolution 3D reconstruction, which requires the collection of large number of micrographs, especially for small molecules, often in the order of thousands. To overcome this barrier, we have analysed each of the distinct conformers with DynaMight. Here, we have successfully characterised the open and closed conformation of the vaccine candidate TcPOP through a synergistic combination of in silico, structural, and in solution techniques. We propose expanding this methodology for regular investigations of small, secreted proteins of sub-80kDa size, especially when recalcitrant to crystallography approaches and where alternating conformations in solution are suspected. The addition of tilted data collection can also be beneficial to alleviate preferential orientation and, therefore, reveal more structural details.

We envisage the possibility to extend this approach to other targets, allowing prediction of the number of 3D classes in vitreous states using cryo-EM data based on MD simulations. In addition, it will promote understanding of conformational heterogeneity at the early stages of cryo-EM data processing and therefore, potentially aid future software developments towards this goal. We provide experimental evidence on the distinct open and closed conformation, which will be invaluable to determine which regions of TcPOP (and potentially other members of prolyl oligopeptidase family) should be targeted to block the enzyme in either conformation, therefore aiding the development of much-needed anti-parasitic therapeutic agents.

Additionally, we characterised the anti-TcPOP polyclonal response, which revealed striking neutralising properties with subsequent rapid lysis of trypomastigote. Furthermore, we isolated one monoclonal antibody, referred to here as mAb1, which also exhibits neutralising activity, likely via a distinct mechanism. While direct binding has not been structurally mapped or demonstrated, we provide strong evidence that mAb1 binds to TcPOP in vitro, further supported by immunofluorescence assays on whole parasite experiments. In the longer term, our findings will allow a structure-guided development of a Chagas vaccine prototype, which is desperately needed in the fight against this major neglected disease.

## Methods

### Bioinformatic analysis of TcPOP and TcPOP homologues
Protein sequences were extracted from the Uniprot database[36] for TcPOP (Q71MD6), TbPOP (Q38AG2), LiPOP (A4ICB5), and HuPOP (P48147) and aligned in BLASTP[37] to retrieve the top 100 homologues of TcPOP, and a phylogenetic tree was built in iTOL[38] after MSA generation in MUSCLE[39]. Comparative homology models were obtained with SWISS-MODEL[40], and when it became available to the scientific community, AlphaFold2 was used to obtain AI-based models[41]. Phylogenetic analysis and sequence conservation mapping on TcPOP were done using CONSURF[42]. The topology diagram of TcPOP was created using PDBsum[43]. Geometry of the models was assessed with MOLPROBITY[44], and 3D alignment performed in PyMOL[45].

### Expression and purification of POPs by bacterial fermentation
The codon-optimised genes encoding for TcPOP and orthologues were purchased from TWIST and inserted in the pET28(a)+ vector (Novagen). The resulting N-ter and C-ter His$_6$-tagged proteins were all recombinantly expressed and purified from the *E. coli* NiCo21(DE3) strain (NEB). Cells were grown at 37 °C supplemented with 2XYT medium (Melford) until OD$_{600nm}$ at 0.6−0.8 was reached. Protein expression was induced using 0.5 mM 1-thio-β-D-galactopyranoside (IPTG, Generon) using an 8 L in-situ bioreactor (INFORS−Techfors S) at 100 rpm, for 18 h at 20 °C with pH control at 7.0 (+/−0.1), resulting in 16 grams of bacterial cell pellet. Lysis was performed using BugBuster (Nalgene) and lysate was clarified by spinning at 50,000 × g for 30 min at 4 °C. Affinity chromatography was performed using Co$^{+2}$-NTA resin (Thermo Fisher). Washing buffer (50 mM sodium phosphate, 500 mM NaCl, 30 mM imidazole, pH 7.4) and elution buffer (50 mM sodium phosphate, 500 mM NaCl, 500 mM imidazole, pH 7.4) were used during affinity purification, followed by dialysis with 3.5 kDa MWCO Dialysis membranes (Thermo Fisher) at 4 °C for 18 h against 20 mM

Hepes and 150 mM NaCl, pH 7.4. Finally, the proteins were concentrated using 30 kDa MWCO Amicon (Millipore) ultra-centrifugal filters to 10 mg/mL and immediately injected into gel filtration column S200 10/300 (Cytiva) equilibrated with 20 mM Hepes and 150 mM NaCl, pH 7.4. Elution fractions containing protein were pooled for further characterisation.

## Western blot analysis

Samples were assessed on the SDS-PAGE gel (Bolt™ Bis-Tris Plus Mini Protein Gels, 4–12% gradient) (Page Ruler Pre-stained Ladder, ThermoFisher) and transferred to nitrocellulose membrane using the Trans-Blot Turbo system (Biorad). Nitrocellulose membrane was incubated with His$_6$-tagged Monoclonal Antibody - HRP (Invitrogen) with 1:3000 v/v dilution in PBS-T, followed by the signal detection using SuperSignal West Pico PLUS Chemiluminescent Substrate (Cat. 34580, ThermoFisher) in an iBright system (Thermo Fisher).

## Differential scanning fluorimetry

Stability measurement of prolyl oligopeptidases was carried out using a fluorescence-assisted thermal unfolding assay. A fluorescence stain SYPRO Orange dye solution (ThermoFisher), was used and diluted to 5x concentration in 20 mM Hepes and 150 mM NaCl, pH 7.4. The assay was performed with a final protein concentration of 1 μM in a total volume of 20 μL. The temperature of the protein samples was gradually increased from 25 °C to 95 °C at a rate of 5 °C per min, using the Rotor-Gene Q (Qiagen). Lastly, the data were analysed using non-linear regression method to determine the midpoint temperatures ($T_m$) of the thermal shift.

## Enzymatic tests for POPs

Fluorogenic POP substrate Suc-Gly-Pro-Leu-Gly-Pro-7-amido-4-methylcoumarin (AMC) (HANGZHOU JHECHEM CO LTD) was diluted in PBS to 1–25 μM final concentrations in 96-well plates (ThermoFisher) to measure enzyme activity parameters in real time, with end-point reactions at 10 and 30 min, using a Gen5™ Microplate Reader and Imager Software. Stock solutions were made in PBS and stored in aliquots at −20 °C. The microplate readings were performed at excitation 360/40 nm and emission 460/40 nm using BioTek Synergy LX Multimode plate reader.

## Mice immunisations

The conducted animal research strictly conformed to the standards delineated by the Federation of European Laboratory Animal Science Associations (FELASA). Ethical clearance for the experimental methodologies was granted by the Danish Animal Experiment Inspectorate, as indicated by their approval number 2018-15-0201-01541. Female BALB/c ByJR mice, aged six weeks, were obtained from Janvier Labs. Mice were kept in individually ventilated cages (IVC). The environmental parameters in the area was set to 8–10 air changes per hour, 22 °C (±2 °C), 55% (±10 %) humidity, and a dark/light cycle of 12 h/12 h. The following vaccine regiment was selected based on previous successful immunisations for the generation of monoclonal antibodies from mouse hybridomas[46,47]. Three mice were immunised intramuscularly with 20 μg TcPOP, emulsified in 50% v/v AddaVax adjuvant (InvivoGen). This was followed by two additional intramuscular injections at biweekly intervals. A concluding intraperitoneal injection of 20 μg TcPOP in PBS was carried out two weeks post the last boost. Three days after the final injection, the mice were humanely euthanized for the extraction of spleen and blood from which sera were subsequently obtained.

## Production, purification and conjugation of anti-TcPOP mAbs

After picking the best immune responders, hybridoma cell lines were generated by fusing splenocytes of immunised mice with myeloma cells (ClonaCell-HY hybridoma cloning kit)[46]. Hybridoma cell lines were harvested 14 days after the fusion and plated into 96-well culture plates in HT supplemented media. Screening of hybridoma cell lines producing antibodies specific to TcPOP was performed by ELISA, as described below. Monoclonal TcPOP-specific hybridoma cell lines were obtained by single cell sorting using the FACSMelody (BD). For large-scale mAb production, hybridoma cell lines were cultured in 4×250 mL cell culture flasks (Corning) as per manufacturer's instructions. Monoclonal antibodies were purified by affinity chromatography using a 5 mL protein G sepharose column (Cytiva) on an ÄKTAexpress system (Cytiva). Antibodies were eluted at 0.8 mg/mL in 0.1 M glycine buffer, pH 2.8, and immediately neutralised with 1:10 v/v 1 M Trizma hydrochloride solution (Sigma Aldrich), pH 9.0 to obtain the final pH 7.4. Buffer exchange to 1 × PBS was performed using a desalting column (Generon), and eluted protein was concentrated with 30 kDa MWCO Amicon (Millipore) to 20 mg/mL. The 3 mAbs were specifically conjugated to HRP using an EZ-Link™ Plus Activated Peroxidase Kit (ThermoFisher) and detected with ECL (Pierce) using the iBright system.

## Enzyme-Linked immunosorbent assay (ELISA)

Antibody-producing hybridoma cells against TcPOP were ascertained through Enzyme-Linked Immunosorbent Assay (ELISA). In summary, MaxiSorp flat-bottom 96-well ELISA plates (ThermoFisher) were coated with recombinant TcPOP (2 μg/mL in PBS) overnight at 4 °C under shaking. Plates were there washed with PBS supplemented with 0.05% v/v Tween20 (PBS-T) and blocking performed for 1 h with casein blocking solution (Pierce). The blocking solution was then removed and replaced with 50 μL of hybridoma supernatant and incubated for 1 h at RT under shaking. Plates were washed three times with PBS-T before undergoing incubation with 1:10000 v/v anti-mouse IgG (γ-chain specific) for 1 h, followed by three 5 min washes with PBS-T. The positive wells were identified by adding TMB plus2 (Kementec) for 20 min and quenched using 0.2 N sulphuric acid. Colorimetric and absorbance signals were measured at 450 nm. Data were analysed using GraphPad (Prism).

## Biological studies

**Cell and parasite culture.** COLO-N680 cells (human oesophageal squamous cell carcinoma line) were maintained in complete MEM medium, consisting of Minimal Essential Medium (Sigma) supplemented with 5% (v/v) heat-inactivated foetal bovine serum (hiFBS, Cytiva), 100 U/ml penicillin, and 100 μg/ml streptomycin. Cells were incubated at 37 °C in a 5% CO$_2$ atmosphere and sub-cultured every 3 days at a 1:5 ratio. COLO-N680 cells were seeded in complete MEM at 80–90% confluency in T25 vented flasks and infected with 2 × 10$^6$ Tissue Culture Trypomastigotes (TCTs) derived from previously infected cells. Infected cultures were maintained for 5 days post-infection. Free-swimming TCTs were isolated by collecting and centrifuging the culture medium at 1600 × g. The resulting pellets were resuspended in Dulbecco's Modified Eagle Medium (DMEM) containing 5% hiFBS and maintained at 37 °C for up to 4 h before use. Motile trypomastigotes were quantified using a hemacytometer.

## In vitro neutralisation cell-based invasion assay

Trypomastigotes obtained as above were incubated in plain DMEM for at least 30 min before starting the assays. Then parasites were pelleted and incubated in DMEM prepared with the sera or monoclonal antibody suspension, according to the concentration assayed. They were incubated for 4 h at 37 °C in an orbital shaker, unless specified differently. Then parasites were washed twice in DMEM to remove the sera/antibodies from the medium and resuspended in DMEM prior to infection. Cells were infected at a multiplicity of infection of 10:1 (parasite:cell). After 4 h of infection, cells were washed three times with PBS to remove non-internalised parasites and incubated with fresh complete MEM for 72 h to allow intracellular amastigote replication. Then, cells were stained with 5 μg/mL of Hoechst and live images were

acquired using an inverted Nikon Eclipse T2i epifluorescence microscope. Infected cells were detached using 1 mL TrypLE Express (Gibco™) for 12 min at 37 °C and fixed with 4% paraformaldehyde (PFA) for 1 h. Cells were then washed in PBS by centrifugation and resuspended in flow cytometry staining buffer (FCSB) and flowed in an Attune NxT Flow Cytometer (Thermo Fisher). Gating was performed using a non-infected culture as a control.

#### Immunofluorescence assays
For the live binding assays, parasites were immediately fixed with 4% (v/v) formaldehyde for 1 h at room temperature following the indicated incubation time with antisera or monoclonal antibody suspensions. Fixed parasites were washed twice with phosphate-buffered saline (PBS) and allowed to settle by gravity onto 10-well slides for immunofluorescence (MP Biomedicals™). Parasites were then permeabilized and blocked in 0.5% saponin containing 10% donkey serum in PBS for a minimum of 1 h, followed by three washes with PBS, each for 5 min. For classic immunostaining, where parasites were directly fixed after isolation from culture, an additional incubation step was performed: parasites were incubated at room temperature for 3 h with either a 1:50 dilution of polyclonal mouse antiserum or a suspension of monoclonal antibody at 1000 µg/mL in 0.5% saponin containing 1% donkey serum in PBS. After three washes with PBS, each for 5 min with agitation, parasites were incubated with a secondary goat anti-mouse antibody (Alexa Fluor 488, Invitrogen) for 1 h at room temperature, protected from light. Slides underwent a final series of three washes with PBS, each for 5 min, and were then allowed to air-dry. Finally, slides were mounted using Vectashield containing DAPI for DNA staining. Images were acquired using an inverted Nikon Eclipse T2i epifluorescence microscope and processed using the NIS-Nikon software.

#### Biophysical studies
**Binding kinetics of TcPOP and anti-TcPOP mAbs using biolayer interferometry.** Binding of anti-TcPOP mAbs to TcPOP was measured by kinetic experiments carried out on an Octet R4 (Sartorius). All samples were buffer exchanged into Sartorius Kinetics Buffer, according to the manufacturer's instructions. All measurements were performed at 200 µL per well in Sartorius kinetic buffer at 25 °C in 96-well black plates (Greiner Bio-One, Cat.No655209). ProG (Cat. Nos. 18-5082, 18-5083, 18-5084) were used to immobilise anti-TcPOP mAbs for 1800 s. Immunogens were four-fold serially diluted in kinetic buffer in the range of 64 nM to 4 nM. Assays were performed in three sequential steps with Octet BLI Discovery 12.2.2.20 software (Sartorius): Step 1, biosensor hydration and equilibration (300 s); Step 2, immobilisation of anti-TcPOP IgG1 mAbs on a ProG biosensor (600 s); Step 3, wash and establish baseline (60 s); Step 4, measure TcPOP association kinetics (1800 s); and Step 5, measure TcPOP dissociation kinetics (600 s). The acquired raw data for the binding of anti-TcPOP mAbs with TcPOP were processed and globally fitted to a 1:1 binding model. Binding kinetics measurements were conducted in triplicate, and reported values represent the average. Data were analysed using Octet Analysis Studio 12.2.2.26 Software (Sartorius) and graphs produced using Graph-Pad (Prism).

#### Epitope binning studies using Bio-layer interferometry (BLI)
Anti-TcPOP mAb2 and anti-TcPOP mAb3 were applied sequentially to assess competition using the BLI. Assays were performed in seven sequential steps with Octet® BLI Discovery 12.2.2.20 software (Sartorius): Step 1, biosensor hydration and equilibration (300 s); Step 2, immobilisation of TcPOP NiNTA biosensors (600 s); Step 3, wash and establish baseline (60 s); Step 4, measure anti-TcPOP mAb2 association

kinetics (1800s); Step 5, measure anti-TcPOP mAb2 dissociation kinetics (600 s); Step 6, measure anti-TcPOP mAb3 association kinetics (1800s); and Step 7, measure anti-TcPOP mAb3 dissociation kinetics (600 s). The acquired raw data for the binding of anti-TcPOP mAbs with TcPOP were processed and globally fitted to a 1:1 binding model with Octet Analysis Studio 12.2.2.26 Software (Sartorius). The binding kinetics measurements were carried out in three replicates. Values reported are the average among triplicates.

#### Estimation of apo-TcPOP using SAXS
SAXS was performed at the B21 beamline (Diamond Light Source, Oxon, UK). TcPOP was buffer exchanged into 20 mM HEPES and 150 mM NaCl (pH 7.4) at 277 K before data collection. Using an Agilent 1200 HPLC system, 50 µL of TcPOP at 8 mg/mL was loaded onto a superdex S200 3.2/300 column for SEC-SAXS. Also, static-SAXS was performed separately at different concentrations extrapolated to zero (from 6 mg/ml to 0 mg/ml, i.e., buffer condition). For SEC-SAXS, X-ray intensity data were collected as the eluent moved from the column to the beam at a flow rate of 0.16 mL/min to collect 600 frames at 3 s intervals, while static SAXS samples were exposed to X-rays to collect 21 frames at 1 s time intervals. The intensity was plotted against its angular dependants q ($q = 4\pi\sin\theta/\lambda$) while, the system operated with an exposure time of 3 s at 12.4 keV (1 Å) using a EIGER 4 M detector. Data were analysed using the BioXTAS RAW[48], ATSAS programme suites[49], DENNS[50], and plotted using GNOM[51].

#### Mass photometry
**Buffer optimisation for TcPOP.** Mass photometry (MP) experiments were conducted using the Refeyn OneMP mass photometer, after cleaning coverslips and gaskets with 100% isopropanol and water. Measurements, performed in triplicate, involved systematic optimisations of pH values in the range 7.0–8.0 and with NaCl concentration range 50–300 mM within a 20 mM BTP buffer. Protein was diluted in buffer to a final concentration of 120 nM into a gasket well, followed by focal point acquisition and data analysis using Refeyn AcquireMP 2.3.1 software. MP movies (6000 frames, 20 frames per sec) were captured within a $10.8 \times 10.8$ µm field and processed with Refeyn DiscoverMP 2.3.0 software. Robust data analysis ensued, leveraging a contrast-to-mass (C2M) calibration approach. Calibration involved introducing 3 µL of a 1:100 v/v pre-diluted NativeMark standard (LC0725, Thermo Scientific) to an acquisition well, yielding masses (66, 146, 480, 1048 kDa) that informed the calibration curve employed in DiscoverMP software. Experiments were performed at the Leicester Institute for Structural and Chemical Biology (LISCB, University of Leicester, UK).

#### Anti-TcPOP binding measurements
MP measurements were conducted using a Refeyn TwoMP (Refeyn Ltd) as previously described in ref. 52. Briefly, glass coverslips (High Precision No. 1.5H, Marienfeld Superior) were cleaned by sequential sonication with Milli-Q H$_2$O, 50% isopropanol, and again Milli-Q H$_2$O. Cleaned coverslips were dried using nitrogen flow. CultureWell™ reusable gaskets (3 mm diameter × 1 mm depth, Grace Bio-Labs) were used to assemble sample chambers. Coverslips were placed on the MP sample stage, and a single gasket was filled with 20 µL DPBS (wo/ calcium and magnesium, pH 7.4, ThermoFisher Scientific) to find focus. TcPOP, mAb1, mAb2, and mAb3 were measured separately at a final concentration of 20 nM. For TcPOP-antibody binding assays, 5 µM TcPOP was mixed with 5 µM mAb2 or mAb3 in a 1.5 mL Eppendorf tube at a 1:1 v/v ratio. The sample mixture was equilibrated for 10 min and diluted 1:100 before data acquisition. Acquisition settings were adjusted within AcquireMP (2023 R1.1, Refeyn Ltd) as a large field of view, frame binning = 2, frame rate = 128.2 Hz, pixel binning = 6, exposure time = 7.65 ms. Movies were taken over 60 s. Mass calibration was performed using an in-house protein standard including

90–720 kDa oligomers. Data were analysed and histograms were created with Discover MP (v2023 R1.2, Refeyn Ltd). Experiments were performed at the New Biochemistry building (University of Oxford, UK).

## MD simulations

Atomics coordinates of TcPOP were retrieved from the AlphaFold database. To calculate the conformational dynamics of TcPOP, all-atom molecular dynamics simulations were conducted on an Ada High Performance Computer (HPC, University of Nottingham) using the GROMACS 2021.2-fosscuda-2020b package. GROMOS 54a7 forefield was applied, and hydrogen atoms were incorporated using the pdb2gmx module, and topology files were generated under periodic boundary conditions (PBC) employing a cubic periodic cell. The protein was centrally placed, solvated using simple point charge (SPC) 216 water molecules and positioned 1 nm from the edges, with NaCl counter ions added for system neutralisation.

Following energy minimisation, the canonical ensemble (NVT) underwent equilibration for 100 ps without pressure coupling, and Berendsen thermostat was initially applied. Subsequently, temperature (298 K) was maintained by velocity rescaling with a stochastic term, while the isothermal-isobaric ensemble (NPT) with a 1 bar pressure for 100 ps, using the Parrinello–Rahman method, was implemented. The LINCS algorithm constrained H-bonds, and the MD simulations ran for 500 ns with a 2-fs time step. The resulting trajectory was analysed using inbuilt functions of the GROMACS package[53].

## Cryo-EM sample preparation, data collection and processing

**Cryo-EM grid preparation.** Homogeneous samples from SEC purified in 20 mM HEPES, 150 mM NaCl, pH 7.4, were freshly used to prepare the grids. Fraction corresponding to the SEC peak at 15 mL (Fig. S2) was used at a final concentration of 0.2 mg/mL. Firstly, cryoEM grids, R1.2/1.3 carbon, Au 300 (Quantifoil), were glow discharged in the presence of amylamine for 30 s at 10 mA on a Quorum GloQube glow-discharge unit. Four microliters of the freshly prepared TcPOP sample were applied to the grid and blotted for 3 s, with blot force 10, prior to flash-cooling in liquid ethane using a Vitrobot Mark IV (FEI ThermoFisher), set at 4 °C and 100% humidity.

## CryoEM data collection

CryoEM grids were imaged using a 300 KeV Titan Krios G3 (ThermoFisher Scientific) transmission electron microscope (Midlands Regional Cryo-EM Facility, University of Leicester) at a calibrated pixel size of 0.656 Å. Electron micrographs were recorded using a K3/GIF (Gatan Imaging Filter) direct electron detector (Gatan Inc.) and EPU automated data acquisition software (ThermoFisher Scientific). Micrograph movies were recorded with 75 fractions, in super resolution, binned by 2 and a total dose of 77 e⁻/pix (dose rate of 15 e-/pix/s), To improve the distribution of particle views, data were collected at 0°, 30° and 35° tilt angle. At 0° tilt (Dataset 1), the defocus range was collected between −2.3 and −0.8 μm, in regular intervals. At 30° and 35° tilt (Dataset 2), the defocus was set to −1.2 μm.

## Cryo-EM image processing

Image processing was carried out on the cryo-EM computational cluster at the University of Leicester. Micrographs were pre-processed using Relion5-beta,[54] sample motion during acquisition was corrected using RELION's own implementation, Dataset 2 was dose-weighted, Dataset 1 was not. Micrographs were then CTF estimated using CTFFIND4.1[55]. Initially each dataset (i.e., 0°, 30° and 35°) was processed separately. In brief, particles were picked using Topaz[56], then extracted with a box size of 256 pixels (corresponding to 168 Å) binned by 4 to 2.624 Å/pix. Particles were filtered in 2D to produce a set of images which were then used for ab-initio model generation in RELION 5-beta. Analysis of the 3D initial models revealed "open" and "closed"

conformations. The full set of Topaz picked particles was classified in 3D with Blush regularisation[54] and the classes were inspected and subcategorised into open and closed particles. Re-extraction of the particles at 1.312 Å/pix followed by 3D auto-refinement and CTF parameter refinement of all three datasets was then carried out. Despite having a 3DFSC sphericity > 0.9, The 0° tilt dataset showed severe preferential orientation and concomitant overestimation of resolution and B-factor parameters was observed. All three datasets were then merged, but this resulted in only marginally improved maps.

Further analysis of the 35° tilt dataset was then performed with DynaMight[57]. This showed that both the open and closed conformations had a moderate degree of continuous motion, explaining why the map quality was limited despite having a large number of high-quality particles. The consensus deformed back projection of both open and closed conformations was then calculated. Map sharpening was done with Relion's automatic B-factor estimation as well as DeepEMhancer[58].

## Cryo-EM model fitting and refinement

The coordinates of the open and closed conformations of TcPOP were docked in the respective cryo-EM maps using phenix.*dock_in_map*[59]. Both maps were refined in PHENIX using real space refinement with 3 cycles of simulated annealing (SA), followed by re-build in place and 10 cycles of refinement at each SA step. Model building was performed in COOT[60]. Structures were validated using the Molprobity tool, as implemented in PHENIX. ChimeraX[61] was used to generate visual molecular graphics. The FSC of the closed and open CryoEM maps was calculated using EBML-EBI FSC-server[62]. Cryo-EM reconstructions have been deposited in the EM Data Bank (EMDB). Data collection and refinement statistics are reported in Supplementary Table Data 1.

## Plasmonic optical nanotweezers: samples preparation and data collection

We used a plasmonic optical tweezers setup, which is a modified modular optical tweezers system (OTKB/M, Thorlabs) in the Advanced Optics and Photonics Lab at Nottingham Trent University (Nottingham, UK) with a 852 nm Fabry-Perot laser diode (FPL852S, Thorlabs)[63]. The laser beam was polarised perpendicular to the centre-to-centre line of two circles of the double nanohole (DNH) structure by using a polariser and a half-wave plate and was collimated and focused on the DNH by a 60X air objective (NA 0.85, Nikon). All trappings were performed at a laser power of 25 mW[64] The transmitted laser intensity was then converted to a voltage signal via a silicon avalanche photodiode (APD120A, Thorlabs) and recorded by a data acquisition card with a sampling rate of 1 MHz. The recorded voltage data (transmission traces) were normalised and filtered using in-house MATLAB scripts, which included a zero-phase Gaussian low-pass filter (MATLAB *filtfilt.m*) with cut-off frequencies of 10 kHz, 1 kHz, 100 Hz, and 10 Hz. Probability density function (PDF) was calculated by using MATLAB function *ksdensity.m*. More information on this specific set-up has been previously reported[65]. SEC Purified TcPOP was used at 1 μM concentration in PBS. DNHs were sealed into the flow cell using cover glass with a double-sided tape as a spacer, providing a microfluidic channel with 3.5 μL volume. The solution was delivered to the flow cell using a 12-way valve and a syringe pump. Initially, TcPOP was infused in the chamber to achieve trapping followed by the sequential infusions of 1 μM, 10 μM, and 100 μM substrate Suc-Gly-Pro-Leu-Gly-Pro-AMC to the flow cell at a flow rate of 2 μL/min.

## Hydrogen-deuterium exchange (HDX) mass spectrometry (MS)

Prior to conducting HDX-MS experiments, peptides were identified by digesting TcPOP using the same protocol and identical liquid chromatographic (LC) gradient as detailed below and performing MS[E] analysis with a Synapt G2-Si mass spectrometer (Waters), applying collision energy ramping from 20 to 30 kV. Sodium iodide was used for calibration, and leucine enkephalin was applied for mass accuracy

correction. MS$^E$ runs were analysed with ProteinLynx Global Server (PLGS) 3.0 (Waters) and peptides identified in 3 out of 4 runs, with at least 0.2 fragments per amino acid (at least 2 fragments in total) and at least 1 consecutive product, with mass error below 7 ppm were selected in DynamX 3.0 (Waters).

For the continuous deuterium labelling, TcPOP (48 μM) was diluted 1:50 v/v in a deuterated buffer at 20 mM HEPES, 150 mM NaCl (96.5% D$_2$O fraction, pH$_{read}$ 7.0), and the exchange reaction was conducted for 2 s, 10 s, 100 s, 1000 s, 10,000 s, and 18 h at room temperature. The exchange reactions were quenched by a 1:1 v/v dilution into ice-cold 100 mM phosphate buffer containing 3 M urea and 70 mM tris(2-carboxyethyl) phosphine (TCEP) (final pH$_{read}$ 2.3). Samples were held on ice for 30 s, snap-frozen in liquid nitrogen, and kept frozen at −80 °C until LC-MS analysis. A maximally labelled sample (MaxD) was produced by labelling TcPOP with 3 M fully deuterated urea in D$_2$O and 2.5 mM TCEP, resulting in a final deuterium content as for the other labelled samples. The maximally labelled sample was quenched after 16 h by 1:1 v/v dilution into ice-cold 100 mM phosphate buffer (final pH$_{read}$ 2.3), held for 30 sec on ice, snap-frozen in liquid nitrogen, and kept frozen at −80 °C until LC-MS analysis. A pulse labelling experiment (two technical replicates) was conducted to assess the protein stability and to rule out irreversible unfolding of the protein over time. An aliquot of TcPOP was quickly thawed and immediately labelled for 10 sec, then exposed to room temperature on the bench for 10,000 s (as under conditions of continuous labelling) and labelled again for 10 s. A summary of the technical details of the HDX-MS data sets is reported in Table S2. Frozen protein samples were quickly thawed and injected into an Acquity UPLC M-Class System with HDX Technology (Waters). The protein was on-line digested at 20 °C into a home-made pepsin column and trapped/desalted with solvent A (0.23% formic acid in water, pH 2.5) for 3 min at 200 μL/min and at 1 °C through an Acquity BEH C18 VanGuard pre-column (1.7 μm, 2.1 mm × 5 mm, Waters). Peptides were eluted into an Acquity UPLC BEH C18 analytical column (1.7 μm, 2.1 mm × 100 mm, Waters) with a 7-min linear gradient raising from 8 to 35% of solvent B (0.23% formic acid in acetonitrile) at a flow rate of 40 μL/min and at 1 °C. Then, peptides went through electrospray ionisation in positive mode and underwent MS analysis with ion mobility separation.

Peptide level deuterium uptake was calculated with DynamX 3.0 (Waters), and datawere visually inspected and analysed. Selected peptides exhibiting evident bimodal HDX behaviour (a clear sign of correlated exchange) were analysed by HX-Express3[31,66] (HX-Express Software). Binomial fitting was applied with optimised fits for the number of amides, and undeuterated mass envelopes were calculated from the peptide sequence and fitted into the experimental undeuterated mass envelope to check the agreement. Subsequently, bimodal deconvolution (double binomial) was enabled for mass envelopes flagged as bimodal. The relative deuterium uptake (Da) and population fraction of both low- and high-mass envelopes were calculated. The fraction of the high-mass population (*y*) over the time points studied (*x*) was subjected to single or double exponential fitting by HX-Express3 to calculate the kinetic parameter Kop (rate of opening), according to the following exponential function in Eq. 1:

$$y = start\ fraction + \left[weight\ fraction \times (1 - exp^{-Kop \times x})\right] \quad (1)$$

To determine the number of backbone amide hydrogens undergoing correlated exchange (#NHs) for the individual peptides, the time point showing maximal difference in HDX between the low- and high-mass population (Max ΔHDX) was identified. Then, the Max ΔHDX value was normalised by the peptide MaxD uptake and corrected for the number of exchangeable amides (N) and D$_2$O fraction (=0.9457),

according to the following Eq. 2, as previously described[67,68]:

$$NHs = \frac{Max\Delta HDX}{MaxDuptake} \times N \times D_{fraction^i} \quad (2)$$

## Reporting summary

Further information on research design is available in the Nature Portfolio Reporting Summary linked to this article.

## Data availability

The data generated in this study have been deposited as coordinate files of the cryo-EM structures of closed and open conformation are available from the PDB with accession codes 9HJI and 9HJJ respectively and 47487042 and 47487043 as raw data IDs in EMPIAR[69]. Static SAXS data have been deposited as SASDW23, whilst SEC-SAXS data were deposited as SASDW33. The plasmonic optical tweezer data have been deposited in the Zenodo database[70] as well as the molecular dynamic trajectory from GROMACS[71]. The raw data relative to the graphs presented in this manuscript are contained in the Source Data file. Source data are provided with this paper.

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

## Acknowledgements

We would like to acknowledge: Dr Nathan Cowieson and Dr Katsuaki Inoue for assisting data collection at B21 beamline (Diamond Light Source), Dr David Staunton (University of Oxford) for data collection of SEC-MALS, Dr Lei Xu (NTU) and Prof. Rahmani Mohsen (NTU) for advice on data analysis of plasmonic optical tweezers experiments. Additionally, we would like to thank, Anu Itansanmi-Ogundayomi (Federal University of Technology Akure, Nigeria), Dr Jody Winter (NTU) and Dr Richard Cowan (University of Leicester). We would like to acknowledge Dr Maria Bassi and Kasper H Björnsson (University of Copenhagen) for advising in antibody methodologies, Dr David Owens (electron Bio-Imaging Centre, Oxon, UK) to provide advice in cryo-EM data quality assessment during the review process and Richard Atherton (LSHTM) for assistance with parasite cultures. The project was funded by: Wellcome grant 204801/Z/16/Z (IC), Royal Society grant IES\R2\232167 (IC). SB is supported by the Nottingham Trent Doctoral School studentship and LB is funded by Novo Nordisk Foundation (NNF170C0026778). FO contribution was supported by Plan Propio of the University of Granada Research Stimulation grant (PP2023.PRI.I.14). We acknowledge The Midlands Regional CryoEM Facility at the Leicester Institute of Structural and Chemical Biology (LISCB), major funding from MRC (MC_PC_17136).

## Author contributions

S.B. expressed and purified proteins, characterised them biophysically and in silico and determined the cryo-EM structure. S.B. and C.L. prepared the samples for cryo-EM data acquisition. S.B., T.J.R., and E.L.H. processed and analysed cryo-EM data. S.B. and A.M.F. isolated and purified antibodies. L.B. immunised mice and performed FACS analysis. S.B., M.K. and W.B.S performed mass photometry experiments and data analysis. V.C. designed, performed and analysed HDX-MS. M.A. and C.Y. performed plasmonic optical tweezers studies. F.O. and J.M.K. performed infection and parasite binding experiments. I.C. conceptualise the experiments and provided funding. S.B. F.O. and I.C. prepared the manuscript and all authors contributed and commented on it.

## Competing interests

All authors declare no competing interests.
