## [Peer Review file · Nature Communications]

Cryo-EM led analysis of open and closed conformations of Chagas vaccine candidate TcPOP

Corresponding Author: Dr Ivan Campeotto

Version 0:

Reviewer comments:

Reviewer #1

(Remarks to the Author)

In this study, Batra *et al.* investigated the structural features of the 80 kDa prolyl oligopeptidase of *Trypanosoma cruzi* (TcPOP), a promising candidate for Chagas vaccine development. They reported the discovery of both open and closed conformations of this protein, whose structures were determined using cryo-EM, as well as multiple other conformations observed through plasmonic optical tweezers. Additionally, they conducted *in vivo* immunization assays to evaluate the immune response against the TcPOP antigen and its homologues, revealing unexpected cross-reactivity.

The study employs a wide array of experimental techniques to characterize TcPOP, which, alongside the immunization experiments, contributes significantly to its value. However, while attempting to resolve the structure of a protein of this molecular weight using cryo-EM is commendable, the discovery of different conformations for TcPOP may not be particularly novel, as similar observations have been reported for homologues. Moreover, there are several major points that need addressing.

Major points:

i) The quality of the PDB models, especially that for the open conformation of TcPOP, is concerning. Although Figure 8D suggests a satisfactory docking of the model into the map with a correlation coefficient of 0.60 (Table 2), the provided wwPDB EM validation report reveals a very poor map-model overlay (section 9.1) with a Q-score of 0.28. Additionally, the provided PDB validation reports correspond to the initial deposition process and are not suitable for review as is clearly stated in the reports. To address these issues, the authors should improve the PDB models and provide a more detailed statistics table for the refinement, including B factors, R.m.s. deviations, MolProbity score, Clashscore, and percentage of poor rotamers.

On the other hand, authors state that both cryo-EM reconstructions and coordinates will be deposited to the EMDB and PDB, respectively. Structural data and models should have been deposited in the appropriate public databases at the time of submission. Otherwise, it is unclear if the structures and the conclusions derived from them are solid enough.

ii) In the SAXS data analysis, the authors exclusively consider the AlphaFold model representing a closed conformation of TcPOP as molecule to dock into the SAXS envelope. However, this approach may overlook the potential existence of alternative conformations, as claim by the authors through their study. Have the authors any explanation to use only the closed conformation? They should show the experimental scattering curve extrapolated to zero-concentration and see if the theoretical scattering profile(s) for the model(s) reproduce(s) well the data (also show the SAXS envelope with the docked model at different orientations is recommended).

In addition, it would be interesting to evaluate the different contributions to the scattering of the open and closed conformations using the program OLIGOMER, as both conformations are expected to exhibit distinct scattering patterns, and explain any preference. Another valuable approach is the utilization of ensemble optimization methods (EOM) to assess the flexibility between the two domains. Guinier plot should be represented in a proper way using the corresponding Guinier limits qR_g and evaluating the linearity of that region.

Additionally, it is recommended to provide a table of the results following the recommendations of the article "2023 update of

template tables for reporting biomolecular structural modelling of small-angle scattering data" published in the journal *Acta Crystallogr D Struct Biol* (PMID: 36762858) and deposit the SAXS data into the SASBDB database for broader accessibility and validation purposes (<https://www.sasbdb.org/>).

iii) I am confused with the MD studies. What does exactly represent the provided Movie 1? Is this movie a full trajectory from this study? Authors need to clarify this and provide an atomic coordinates trajectory of the 500 ns simulation. Were replicas of the simulations made? Could be the open structure observed by MD used as initial model to dock into the cryo-EM maps?

Minor points:

i) Have the authors performed structural predictions of the different conformations of TcPOP using AlphaFold? While the authors state they used the AlphaFold model from Uniprot (Q71MD6) as a starting point for the MD simulations, it could be interesting to generate different predictive models to explore if the different states for TcPOP can be predicted. Additionally, a supplemental figure containing a superposition of different predicted structures, along with the corresponding per-residue model confidence score (pLDDT), is recommended.

ii) To enhance clarity for readers who may not be familiar with these types of proteins, I suggest to the authors include more information in figures with structures, such as labeling the β -propeller domain, $\alpha\beta$ -hydrolase domain, or other related structural features (amino acid positions, etc.).

iii) Please, include cites to refer the software used for the SAXS data analysis and the WHO 2023 report. In addition, the title for SEC-SAXS section in Materials and Methods should be updated.

iv) Error bars should be included in Figure S1B and S1C.

v) Fig. S7 is not referenced.

vi) Please, correct the following typos:

- Legend of Figure 1. Please, include the number 5 for the Homo sapiens POP lane.
- In the Keywords, replace "SAX" by "SAXS".
- Some TcPOP/LiPOP/HuPOP are not properly displayed in italics: Lines 94 and 98 on page 2, Line 166 on page 4, Line 240 on page 5, Figures S2 and S3, Legend of Figure S5, Line 413 on page 9 and Line 524 on page 12.
- Legend of Figure S2. The elution volume should read 14-15 mL; 12 mL for LiPOP. Please replace "expection" by "exception". Also, check elution volume in legend of Figure 1.
- Legend of Figure S4. Please, replace "red" by "pink".
- Please be consistent with the acronyms (Tc80 in Figures 2, 3, S1, and S9).
- 1-Thio- β -D-galactopyranose in Line 119, page 3.
- Enzyme-Linked in Line 186, page 4.
- "nvestigate" in Line 95, page 2.
- Line 410 page 9, please, check the references to the figures.
- "Corelation" in Table 2.
- Letters for the panels "D" and "E" in Figure 8 should read "E" and "F", respectively.

Reviewer #2

(Remarks to the Author)

The paper characterized a vaccine candidate previously described using a lethal model Tc80. The vaccine candidate is presented in both parasite stages in the host: amastigote and trypomastigote. They used a different adjuvant, doubled the amount of protein (10 ug from Binova et al. to 20 ug in the current work), and did a deep characterization of the vaccine candidate. However, I don't see the study's novelty, so I suggest highlighting it in the last part of the introduction and abstract.

Since they are changing the adjuvant, I would like a phrase about the differences (if any) between AddaVax and oligodeoxynucleotides CpG (ODN-CpG). Why do they duplicate the amount of protein? There is a dose range study. I want a side-to-side comparison between the original TcPOP and the new TcPOP formulation. Do you have a better/similar response to the vaccine? What is the advantage of your TcPOP over the last one? Or at least compare similar parameters to know the novelty, or again, is this paper just a characterization of the protein?

For the introduction, please add a brief description of the immune response of the parasite and what is the profile of the vaccine that you should expect. It looks like the focus of this vaccine is the humoral response. So, I would like to see an in-vitro experiment where they use the sera from immunized mice and see the effect of the antibodies during the interaction, internalization, and replication of the parasite at (24-48h). 1. Incubation of parasite with the sera, washing steps and adding the parasite for 1-4 h to the cells, then washing steps and evaluation by qPCR/bioluminescent/fluorescent evaluation of the interaction, internalization, and replication of the parasite at (24-48h).

Add WB, with parasite lysate *T. brucei*, *Leish*, and *T. cruzi*, compared to the anti-TcPOP mice' blood sera and the mAB1-3. Please add microscopy of the *T. cruzi* stained with the best mAB(1-3) in both stages, amas and trypos.

In general, please prepare a better set of figures; some legends are difficult to read.

Reviewer #3

(Remarks to the Author)

This manuscript by Bartra, S., et al., presents the first-time structure of TcPOP by CryoEM in two different conformations: open and close state at reported resolution 3.0 Å and 2.5 Å, respectively. Together with plasmonic optical tweezers techniques, the authors shown that there were multiple conformations of TcPOP that exists in solution. The authors also produced TcPOP antibody from mouse, which has distinct specificity toward parasite species but not human enzyme HuPOP. Although the manuscript presents valuable data, there are some concerns/revisions that may help to improve the manuscript.

Major concerns:

- MATERIALS AND METHODS, Cryo-EM model fitting and refinement (line 332-333): "...Both maps were refined and validated using phenix.real_space_refine": The manuscript describes that docking and refining PDB models were done automatically through Phenix only. Is there a missing step that was not mentioned in these processes from the manuscript?
- RESULTS, Figure 8: Regarding the two reported maps of TcPOP by cryo-EM, although the two maps were reported at 3.0 Å and 2.5 Å, it is hard to detect sidechain features from the two maps in Figure 8A-D. Thus, I wonder if the resolution were overestimated due to preferential orientation problem. I would suggest the authors evaluate their maps by 3DFSC program (Tan, Y.Z., et al., 2017) and showing histogram of directional FSCs of the two maps in supplement file. In addition, authors may consider presenting local resolution of the two maps in supplement file.
- RESULTS, TcPOP structure determination using single particle CryoEM (line 427-438): the two cryo-EM structures were poorly described in the manuscript. It is hard for me to follow without checking other papers to know where is the α/β hydrolase, or β -propeller domain.
- How is the close conformation of TcPOP comparing with other available POP structures? What is the different between open and close conformation of TcPOP? Does the α/β hydrolase change its conformation between the open and close state?
- Why did TcPOP antibody bound specific to others parasite homologous but not human HuPOP? Is there any explanation from structural point of view, sequence analysis, or prediction of the binding site?
- RESULTS, SEC-SAXS (line 407-410): Since there are multiple conformation was observed by Cryo-EM and plasmonic optical tweezers, why only a compact globular conformation was observed by SAXS?
- Line 498 – 499: "The evident stability of the α/β hydrolase domain...": As in the Movie 1, although the α/β hydrolase is more stable than the β -propeller, it seems that there are some changes in the conformation of α/β hydrolase domain, which were not addressed in the RESULTS of the manuscript.

Minor comments:

- Line 411-412 "Mass photometry ... pH values (Fig. D-E)" Which figure should be indicated here? Is it Figure 5? If yes, there is no Figure 5E.
- Line 428 – 430: "The initial data collection ... map was not produced (Fig.S8).": Should it be Fig.S7 here instead of Fig. S8?
- Line 434: "... continuous density and minimal anisotropic features (Fig.S8)": Fig.S8 of the manuscript presents "2D-classification of extracted particles of TcPOP representing distinct multiple conformations". The figure doesn't show the improvement in the number of views.
- Line 497-498: "...overall motion of ~220 between the domains, with the z-axis of rotation centred on Gly 424.", it is not clear for me where is the Gly 424 and the z-axis. Authors may consider indicating it in the Figure 8.
- There are two Figure 8Ds in Figure 8.

References:

Tan, Y.Z., Baldwin, P.R., Davis, J.H., Williamson, J.R., Potter, C.S., Carragher, B. and Lyumkis, D., 2017. Addressing preferred specimen orientation in single-particle cryo-EM through tilting. Nature methods, 14(8), p.793.

Version 1:

Reviewer comments:

Reviewer #1

(Remarks to the Author)

While the authors have addressed the concerns raised in the previous revision, there are still some issues in this new version of the article that need attention, particularly regarding the resolution of the structures. In the wwPDB EM Validation Reports for the first version of the manuscript, the resolutions of the closed and open conformations were 2.5 Å and 3.0 Å, respectively, and these values were consistently reported throughout the manuscript, including in the abstract. However, in this version, the wwPDB EM Validation Reports show resolutions of 3.57 Å (closed) and 3.82 Å (open), while the article indicates 2.8 Å and 3.0 Å for the closed and open conformations, respectively. The authors should clarify this discrepancy or, if it is a mistake, update these values accordingly. If these are indeed the new resolutions, can the authors still claim that this is one of the smallest cases, and at the highest resolution reported, in terms of resolving multiple enzyme conformations by single-particle cryo-EM?

Additionally, in the newly added SAXS table, I noticed a significant discrepancy in the static SAXS data between the R_g

derived from the Guinier (38 Å) and the P(r) analysis (71 Å). Please explain or correct this inconsistency. Finally, it would be helpful if the authors could provide the P(r) plots.

Reviewer #2

(Remarks to the Author)

I appreciate the author's effort in conducting the additional experiments; the manuscript looks significantly improved as a result. I still have the following questions:

Why do they double the amount of protein in the dose range study? What is the rationale behind this?

I would like a side-by-side comparison between the original TcPOP and the new TcPOP formulation. Does the new formulation elicit a better or comparable response to the vaccine?

What is the advantage of this new TcPOP over the previous one? At the very least, could you compare similar parameters to highlight any novelty? Or is this paper primarily focused on protein characterization?

Thank you for including the neutralization assay. However, *T. cruzi* mutants are generally less virulent than the wild type (WT). Could you include the WT for comparison?

I also don't see a specific band for the western blot (WB). This is especially important because the monoclonal antibody is expected to recognize something specific in *T. cruzi*. Here's an example of the expected result with your antigen: DOI: 10.1371/journal.ppat.1012764, Figures S6 and S7.

Reviewer #3

(Remarks to the Author)

The authors' revision has significantly improved the quality of the manuscript and addressed most of the issues I raised in the previous version. However, I have one remaining concern regarding a minor discrepancy with the global resolution of the open and closed TcPOP structures. As shown in the FSC curve from the PDB validation reports, as well as the global FSC curves in Figure S14, the global resolutions for the open and closed TcPOP structures doesn't appear to be 3.0 Å and 2.8 Å with FSC 0.143 cutoff as stated in the manuscript. I also see that the reported resolution by the authors of open and closed TcPOP structures in the PDB validation reports were 3.82 Å and 3.57 Å, respectively. I kindly request that the authors update the manuscript to reflect the resolution values reported in the PDB validation reports.

Version 2:

Reviewer comments:

Reviewer #1

(Remarks to the Author)

The authors have addressed the major points raised, and the manuscript has improved considerably compared to the first version.

A few considerations for the authors to take into account:

- Please ensure accurate citation of data, particularly for references 3 and 5.
- Additionally, check reference 11.
- Clarify the limitations of using static SAXS, as discussed in the authors' rebuttal letter, and ensure they are included in the current version of the manuscript.

Reviewer #2

(Remarks to the Author)

Thank you.

I still want to see the Western blot and the specificity of TcPOP. The assay I'm looking for is similar to what I mentioned in the Versteeg paper. I'd like to load TcPOP protein along with lysates from *T. cruzi* and other parasites like *T. brucei* and *Leishmania*. The Western blot can be performed with and without reducing conditions. Then, I'll use antisera from vaccinated mice to determine if the detection is specific.

Let me know your thoughts!

We would like to thank all the Reviewers for their comments, as they encouraged us to further explore the structural and functional aspects of this project and generate additional novel findings which will move forward the field and our own research in follow-up studies.

Please find our point-to-point responses below.

REVIEWER COMMENTS

Reviewer #1 (Remarks to the Author):

In this study, Batra *et al.* investigated the structural features of the 80 kDa prolyl oligopeptidase of *Trypanosoma cruzi* (*TcPOP*), a promising candidate for Chagas vaccine development. They reported the discovery of both open and closed conformations of this protein, whose structures were determined using cryo-EM, as well as multiple other conformations observed through plasmonic optical tweezers. Additionally, they conducted *in vivo* immunization assays to evaluate the immune response against the *TcPOP* antigen and its homologues, revealing unexpected cross-reactivity.

The study employs a wide array of experimental techniques to characterize *TcPOP*, which, alongside the immunization experiments, contributes significantly to its value. However, while attempting to resolve the structure of a protein of this molecular weight using cryo-EM is commendable, the discovery of different conformations for *TcPOP* may not be particularly novel, as similar observations have been reported for homologues. Moreover, there are several major points that need addressing.

Major points:

i) The quality of the PDB models, especially that for the open conformation of *TcPOP*, is concerning. Although Figure 8D suggests a satisfactory docking of the model into the map with a correlation coefficient of 0.60 (Table 2), the provided wwPDB EM validation report reveals a very poor map-model overlay (section 9.1) with a Q-score of 0.28. Additionally, the provided PDB validation reports correspond to the initial deposition process and are not suitable for review as is clearly stated in the reports. To address these issues, the authors should improve the PDB models and provide a more detailed statistics table for the refinement, including B factors, R.m.s. deviations, MolProbity score, Clashscore, and percentage of poor rotamers.

Alphafold models of TcPOP in either closed or open conformations were docked using PHENIX dock-in-map and morphed into the experimental ab-initio maps in PHENIX refine using 3 cycles of simulated annealing (SA) with 10 refinement cycles at each step to reduce phase memory.

The final Q-scores have improved to 0.38 for closed conformation (PDB code 9HJI) and to 0.36 for open conformation (PDB code 9HJJ) with no outliers in the Ramachandran plot.

Our resolution is in line with previously reported cases:

<https://journals.iucr.org/d/issues/2021/09/00/qr5001/qr5001.pdf>

Also, the contour level of the EM maps corresponds to their respective values from Chimera, are shown below as extract from the validation report.

9.4 Atom inclusion ⓘ

As we are not making claims about map resolution for individual residues, we opted to insert in the main Figure 4a-b two images with the local resolution instead to aid the readers. Please note that such Q values are also reflective of the lack of density for some loops which are disordered, as corroborated by MD *in silico* studies (please refer to the RMSF values). The Material and Methods section in the paper as well as the statistics table were updated accordingly to reflect the new statistics. Figure 4 also includes these new maps overlaid to the coordinates.

More detailed refinement statistics are reported in Table 2.

On the other hand, authors state that both cryo-EM reconstructions and coordinates will be deposited to the EMDB and PDB, respectively. Structural data and models should have been deposited in the appropriate public databases at the time of submission. Otherwise, it is unclear if the structures and the conclusions derived from them are solid enough.

Both closed and open conformation were deposited in EMDB with 9HJI (closed) and 9HJJ (open). The structures are on hold until publication, but the final report has been shared as an attached file to submission.

ii) In the SAXS data analysis, the authors exclusively consider the AlphaFold model representing a closed conformation of TcPOP as molecule to dock into the SAXS envelope. However, this approach may overlook the potential existence of alternative conformations, as claim by the authors through their study. Have the authors any explanation to use only the closed conformation? They should show the experimental scattering curve extrapolated to zero-concentration and see if the theoretical scattering profile(s) for the model(s) reproduce(s) well the data (also show the SAXS envelope with the docked model at different orientations is recommended).

We thank the reviewer for this insight into SAXS data collection. We reprocessed the data, and we were indeed able to deconvolute the data to fit the two different conformations, closed and open, from cryo-EM coordinates. We also performed static SAXS experiments as suggested by the reviewer and were successfully able to extrapolate sample concentration to zero-concentration as requested, provided in Figure S15.

In addition, it would be interesting to evaluate the different contributions to the scattering of the open and closed conformations using the program OLIGOMER, as both conformations are expected to exhibit distinct scattering patterns and explain any preference. Another valuable approach is the utilization of ensemble optimization methods (EOM) to assess the flexibility between the two domains. Guinier plot should be represented in a proper way using the corresponding Guinier limits qR_g and evaluating the linearity of that region.

We followed the reviewer's advice and used DENSS to deconvolute open and close domain motions, which is equivalent to OLIGOMER in its function. DENSS was used instead of OLIGOMER due to better accuracy in heterogenous datasets (DOI: 10.1016/bs.mie.2022.09.018). Furthermore, we performed HDX-MS to provide additional complementary information on the kinetics of transition between open and close conformation, which allowed us to calculate the relative rates at which individual structural elements of TcPOP transition between the closed and open conformation (Fig.

8 and Supplementary Figs. 14-15, Supplementary Table 2).

Additionally, it is recommended to provide a table of the results following the recommendations of the article “2023 update of template tables for reporting biomolecular structural modelling of small-angle scattering data” published in the journal *Acta Crystallogr D Struct Biol* (PMID: 36762858) and deposit the SAXS data into the SASBDB database for broader accessibility and validation purposes (<https://www.sasbdb.org/>).

We followed the reviewer’s comment and added a table for SAXS data for both, SEC-SAXS and static SAXS experiments (Fig. S16), based on the manuscript recommended by the reviewer. This has been added to the references. Both datasets have also been deposited in SASDB with accession numbers SASDW23 and SASDW33.

iii) I am confused with the MD studies. What does exactly represent the provided Movie 1? Is this movie a full trajectory from this study? Authors need to clarify this and provide an atomic coordinates trajectory of the 500 ns simulation. Were replicas of the simulations made? Could be the open structure observed by MD used as initial model to dock into the cryo-EM maps?

Thank you for the comments on our MD simulation. The movie represents a transition between closed and open conformations based on the reported cryo-EM experimental structures. As this is confusing, we removed it. The total file size of 500 ns MD simulation raw data is 138 Gb which exceeds the quota for deposition in public databases such Zenodo, but we could supply a link to the data hosted on our HPC on request. Simulations were run in 1 ps intervals for 500 ns, which avoid running replicas due to its high confidence and precise simulation. Additionally, running replicas would be too computationally expensive and single MD simulations are widely accepted in the MD community such as CCPBIOSIM.

Minor points:

i) Have the authors performed structural predictions of the different conformations of *TcPOP* using AlphaFold? While the authors state they used the AlphaFold model from Uniprot (Q71MD6) as a starting point for the MD simulations, it could be interesting to generate different predictive models to explore if the different states for *TcPOP* can be predicted. Additionally, a supplemental figure containing a superposition of different predicted structures, along with the corresponding per-residue model confidence score (pLDDT), is recommended.

We used AlphaFold3 with a single monomer as input and obtained only the closed conformation. When we used AlphaFold multimer in the attempt of predicting the open conformation, we were able to obtain only an open dimer conformation. However, as this has low confidence, we decided to report it in this rebuttal only. We also added here the prediction of the closed conformation for completeness.

ii) To enhance clarity for readers who may not be familiar with these types of proteins, I suggest to the authors include more information in figures with structures, such as labeling the β -propeller domain, α/β -hydrolase domain, or other related structural features (amino acid positions, etc.).

We agree with this suggestion and added a topology file from PDBsum in the supplementary data (Fig. S11) to aid the reader, as well as indicated specific residue intervals in HDX-MS figures.

iii) Please, include cites to refer the software used for the SAXS data analysis and the WHO 2023 report. In addition, the title for SEC-SAXS section in Materials and Methods should be updated.

This has now been updated

iv) Error bars should be included in Figure S1B and S1C.

This has now been amended

v) Fig. S7 is not referenced.

Figure numbering has changed and this has been considered

vi) Please, correct the following typos:

These changes have been made accordingly

- Legend of Figure 1. Please, include the number 5 for the Homo sapiens POP lane.
- In the Keywords, replace “SAX” by “SAXS”.
- Some TcPOP/LiPOP/HuPOP are not properly displayed in italics: Lines 94 and 98 on page 2, Line 166 on page 4, Line 240 on page 5, Figures S2 and S3, Legend of Figure S5, Line 413 on page 9 and Line 524 on page 12.
- Legend of Figure S2. The elution volume should read 14-15 mL; 12 mL for *LiPOP*. Please replace “exeption” by “exception”. Also, check elution volume in legend of Figure 1.
- Legend of Figure S4. Please, replace “red” by “pink”.
- Please be consistent with the acronyms (Tc80 in Figures 2, 3, S1, and S9).
- in Line 119, page 3.
- Enzyme-Linked in Line 186, page 4.
- “nvestigate” in Line 95, page 2.
- Line 410 page 9, please, check the references to the figures.
- “Corelation” in Table 2.
- Letters for the panels “D” and “E” in Figure 8 should read “E” and “F”, respectively.

Reviewer #2 (Remarks to the Author):

The paper characterized a vaccine candidate previously described using a lethal model Tc80. The vaccine candidate is presented in both parasite stages in the host: amastigote and trypomastigote. They used a different adjuvant, doubled the amount of protein (10 ug from Binova et al. to 20 ug in the current work), and did a deep characterization of the vaccine candidate. However, I don't see the study's novelty, so I suggest highlighting it in the last part of the introduction and abstract. Since they are changing the adjuvant, I would like a phrase about the differences (if any) between AddaVax and oligodeoxynucleotides CpG (ODN-CpG). Why do they duplicate the amount of

protein? There is a dose range study. I want a side-to-side comparison between the original TcPOP and the new TcPOP formulation. Do you have a better/similar response to the vaccine? What is the advantage of your TcPOP over the last one? Or at least compare similar parameters to know the novelty, or again, is this paper just a characterization of the protein?

For the introduction, please add a brief description of the immune response of the parasite and what is the profile of the vaccine that you should expect. It looks like the focus of this vaccine is the humoral response.

The following text has been added in line 99 of current manuscript:

T. cruzi triggers a complex immune response in humans, involving both innate and adaptive systems. Initially, the innate response is activated, including cytokine production by phagocytes and natural killer cells, which produce interferon-gamma¹⁴. The adaptive response involves T and B lymphocytes, with T cells orchestrating cellular immunity and B cells producing antibodies, although these are often ineffective due to the parasite's evasion strategies¹⁵. *T. cruzi* employs mechanisms including hijacking of the TGF- β signalling pathway and modulating immune responses to evade detection and establish a chronic infection. Achieving a potent and protective humoral response would be an ideal approach to block transmission in humans¹⁶.

So, I would like to see an *in vitro* experiment where they use the sera from immunized mice and see the effect of the antibodies during the interaction, internalization, and replication of the parasite at (24-48h). 1. Incubation of parasite with the sera, washing steps and adding the parasite for 1-4 h to the cells, then washing steps and evaluation by qPCR/bioluminescent/fluorescent evaluation of the interaction, internalization, and replication of the parasite at (24-48h).

We thank the reviewer for this insightful suggestion. We have now incorporated a comprehensive set of biological assays that fully meet and exceed the requested requirements. We systematically tested the neutralizing activity of the three antisera obtained after immunizations. Following the reviewer's suggestion, we incubated the trypomastigotes with varying concentrations of sera for 4 hours, then added the parasites to cells, allowing infection to proceed for an additional 4 hours, and monitored infection for a further 72 hours.

To evaluate their activity, we utilized the intrinsic fluorescent properties of our mutant parasites, employing automated readouts as recommended. In cases where some conditions exhibited signals below the detection threshold of fluorescent readers, we also measured infectivity using flow cytometry. We have included confirmatory data by incorporating representative images of live and fixed parasites using real-time and epifluorescence microscopy (Fig. 1-2 and Supplementary Fig. S7,9).

Additionally, we repeated similar procedures to assess the activity of monoclonal antibodies derived from hybridomas produced from the mouse that yielded the antiserum with the best neutralizing activity. Finally, we thoroughly characterized the dynamics and patterns of binding and neutralization of both the best polyclonal serum and monoclonal antibody by conducting multiple assays under various conditions of concentration, temperature, and time, combining again live and epifluorescence microscopy (Fig. 3 and Supplementary Fig. S10).

Add WB, with parasite lysate *T. brucei*, *Leish*, and *T. cruzi*, compared to the anti-TcPOP mice' blood sera and the mAB1-3.

Although, after the comprehensive set of assays we have incorporated, it could be redundant, we have included in here a WB figure in response to the reviewer, using our best monoclonal antibody (which has been HRP-conjugated) against the lysates from *T. cruzi* and the related kinetoplastids (*Trypanosoma brucei* and *Leishmania aethiopica*.) he/she suggested.

Western blot analysis of multiple lysates from *T. cruzi* and related trypanosomatids tested against HRP-conjugated IM1-mAb1 at a 1:5000 v/v dilution. Lane 1: *T. brucei* bloodstream forms; Lane 2: *Leishmania aethiopica* promastigotes; Lane 3: *T. cruzi* trypomastigotes; Lane 4: *T. cruzi* amastigotes.

Please add microscopy of the *T. cruzi* stained with the best mAB(1-3) in both stages, amas and trypos.

We have now incorporated six elements, each with multiple image panels, into the main figures, along with an additional nine elements with multiple panels in four supplementary figures. Together, these elements illustrate the binding patterns observed in both extracellular infective trypomastigotes and excised intracellular replicative amastigotes under various physiological conditions, such as the metabolic states that parasites experience over time during each life-cycle stage. To address issues arising from rapid metabolic variations in this protozoan, we combined live and fixed binding-immunoassays to provide more comprehensive information about the immunoprotective nature of the antibodies derived from TcPOP immunizations.

Additionally, we have produced two supplementary videos that clearly demonstrate the differences in binding profiles between the most effective polyclonal antiserum and the best monoclonal antibody obtained from the same mouse (Supplementary Videos S2-3).

In general, please prepare a better set of figures; some legends are difficult to read.

Following the reviewer's recommendations, we have revised the figures and legends to enhance clarity. Additionally, we have included several schematic representations of the protocol developed to test our biological hypothesis regarding the neutralizing activity of the antisera and monoclonal antibodies. The legends have been expanded to provide a more detailed explanation of the figure contents. We have also extended the supplementary figures to offer a more comprehensive view of the full set of assays conducted.

Reviewer #3 (Remarks to the Author):

This manuscript by Bartra, S., et al., presents the first-time structure of TcPOP by CryoEM in two different conformations: open and close state at reported resolution 3.0 Å and 2.5 Å, respectively. Together with plasmonic optical tweezers techniques, the authors shown that there were multiple conformations of TcPOP that exists in solution. The authors also produced TcPOP antibody from mouse, which has distinct specificity toward parasite species but not human enzyme HuPOP. Although the manuscript presents valuable data, there are some concerns/revisions that may help to improve the manuscript.

Major concerns:

- MATERIALS AND METHODS, Cryo-EM model fitting and refinement (line 332-333): "...Both maps were refined and validated using phenix.real_space_refine": The manuscript describes that docking and refining PDB models were done automatically through Phenix only. Is there a missing step that was not mentioned in these processes from the manuscript?

We expanded on the methods section to include step by step explanation of model fitting onto experimental maps and cycles of simulated annealing and auto-building as implemented in PHENIX (now lines 415-417).

- RESULTS, Figure 8: Regarding the two reported maps of TcPOP by cryo-EM, although the two maps were reported at 3.0 Å and 2.5 Å, it is hard to detect sidechain features from the two maps in Figure 8A-D. Thus, I wonder if the resolution were overestimated due to preferential orientation

problem. I would suggest the authors evaluate their maps by 3DFSC program (Tan, Y.Z., et al., 2017) and showing histogram of directional FSCs of the two maps in supplement file.

We have based the resolution cut-off on the FSC profile as suggested in PHENIX, which is in agreement with the EMBL-EBI server FSC profile, which has also been uploaded in EMDB.

We also reported this here for ease (test123=closed, test987=open):

In addition, authors may consider presenting local resolution of the two maps in supplement file.

We think that this information is crucial and the differences on local resolution are likely to be attributed to intrinsic flexibility of TcPOP enzyme. We therefore decided to show it as recommended and due to the relevance in the context of additional HDX-MS data, we moved it to Figure 4 in the main section of the manuscript.

- RESULTS, TcPOP structure determination using single particle CryoEM (line 427-438): the two cryo-EM structures were poorly described in the manuscript. It is hard for me to follow without checking other papers to know where is the α/β hydrolase, or β -propeller domain.

- How is the close conformation of TcPOP comparing with other available POP structures? What is the different between open and close conformation of TcPOP? Does the α/β hydrolase change its conformation between the open and close state?

The close conformation changes very little across other POP homologues. For the same family, only Porcine Muscle POP (PDB code 1QFM) and *Pyrococcus furiosus* POP (PDB code 5T88) have been solved to date and the r.m.s.d is respectively 2.7 Å and 2.1 Å respectively for the C-alpha carbons. The r.m.s.d. between closed and open cryo-EM conformations is instead 7Å due to inter-domain motions rather than individual changes of conformations of the individual domains.

- Why did TcPOP antibody bound specific to others parasite homologous but not human HuPOP? Is there any explanation from structural point of view, sequence analysis, or prediction of the binding site?

The sequence identity of TcPOP compared to TbPOP and LiPOP is 73% and 63 % respectively, whilst it is 43% for HuPOP. We can only speculate at this stage, that there is more likelihood that such high identity increases the likelihood to share common epitopes, although there could be an evolutionary explanation, which we may investigate in future research.

- RESULTS, SEC-SAXS (line 407-410): Since there are multiple conformation was observed by Cryo-EM and plasmonic optical tweezers, why only a compact globular conformation was observed by SAXS?

- Line 498 – 499: “The evident stability of the α/β hydrolase domain... ”: As in the Movie 1, although the α/β hydrolase is more stable than the β -propeller, it seems that there are some changes in the conformation of α/β hydrolase domain, which were not addressed in the RESULTS of the manuscript.

We would like to thank the reviewer for suggesting to explore whether SAXS data could be deconvoluted. We used DENNS and included the results in Figure 4c. The information has been added in the RESULTS section (lines 629-630).

Minor comments:

- Line 411-412 “Mass photometry ... pH values (Fig. D-E)” Which figure should be indicated here? Is it Figure 5? If yes, there is no Figure 5E.

- Line 428 – 430: “The initial data collection ... map was not produced (Fig.S8).”: Should it be Fig.S7 here instead of Fig. S8?

- Line 434: “... continuous density and minimal anisotropic features (Fig.S8)”: Fig.S8 of the manuscript presents “2D-classification of extracted particles of TcPOP representing distinct multiple conformations”. The figure doesn't show the improvement in the number of views.

- Line 497-498: “...overall motion of ~ 220 between the domains, with the z-axis of rotation centred on Gly 424.”, it is not clear for me where is the Gly 424 and the z-axis. Authors may consider indicating it in the Figure 8.

- There are two Figure 8Ds in Figure 8.

The text has been changed accordingly but please note that Supplementary and Figure numbers have changed due to the new data being reported. The initial choice of Gly 424 as reference residue was simply arbitrary and more detailed experimental information is now provided by HDX-MS, which replaces this information.

References:

Tan, Y.Z., Baldwin, P.R., Davis, J.H., Williamson, J.R., Potter, C.S., Carragher, B. and Lyumkis, D., 2017. Addressing preferred specimen orientation in single-particle cryo-EM through tilting. *Nature methods*, 14(8), p.793.

This reference has been added.

REVIEWERS COMMENTS

Reviewer #1 (Remarks to the Author):

While the authors have addressed the concerns raised in the previous revision, there are still some issues in this new version of the article that need attention, particularly regarding the resolution of the structures. In the wwPDB EM Validation Reports for the first version of the manuscript, the resolutions of the closed and open conformations were 2.5 Å and 3.0 Å, respectively, and these values were consistently reported throughout the manuscript, including in the abstract. However, in this version, the wwPDB EM Validation Reports show resolutions of 3.57 Å (closed) and 3.82 Å (open), while the article indicates 2.8 Å and 3.0 Å for the closed and open conformations, respectively. The authors should clarify this discrepancy or, if it is a mistake, update these values accordingly. If these are indeed the new resolutions, can the authors still claim that this is one of the smallest cases, and at the highest resolution reported, in terms of resolving multiple enzyme conformations by single-particle cryo-EM?

Additionally, in the newly added SAXS table, I noticed a significant discrepancy in the static SAXS data between the R_g derived from the Guinier (38 Å) and the $P(r)$ analysis (71 Å). Please explain or correct this inconsistency. Finally, it would be helpful if the authors could provide the $P(r)$ plots.

We acknowledge the mistake in reporting the **resolution values** and have **now corrected** them throughout the manuscript. The revised resolutions from the wwPDB EM Validation Reports are 3.6 Å (closed conformation) and 3.8 Å (open conformation). The previously reported values (2.8 Å and 3.0 Å) were a legacy error from an earlier refinement stage. We have now ensured that all instances in the text, including the abstract and figure legends, reflect the correct values.

Additionally, we added in the methods and in Table 1 further information (lines 398-402) referencing the usage of DynaMight which has also been included in Figure S13 (Supplementary Data).

Regarding the reviewer's question about whether our claim that this represents one of the smallest cases resolved at high resolution for multiple enzyme conformations remains valid, we have revisited the literature. While the new resolution values are slightly lower than initially reported, they are still within the high-resolution range for single-particle cryo-EM. We have modified the relevant statements in the manuscript to reflect this adjustment while maintaining the significance of our findings:

Lines 749-750 "This is one of the smallest cases, and at the highest resolution reported, in terms of resolving multiple enzyme conformations in a vitreous state by single-particle cryo-EM for this class of enzymes"

We appreciate the reviewer's feedback on the SAXS data. The discrepancy between the R_g values from Guinier (38 Å) and $P(r)$ analysis (71 Å) in the static SAXS measurements is due to the presence of larger multimers that persist even when extrapolating to zero concentration. If the dissociation constant for these species is low, dilution alone does not significantly shift the equilibrium, meaning that the batch SAXS measurements still capture a polydisperse mixture. As a result, extrapolating to zero concentration cannot fully recover the properties of a species that was never measured in isolation. Given these limitations, we produced the $P(r)$ plots but **we opted to omit them as well as the static SAXS data**, which were initially included at the reviewer's request in an earlier revision. Instead, we rely on SEC-SAXS, which directly isolates the monomeric species and provides a more accurate dataset. We believe SEC-SAXS offers the most reliable approach in this case.

We also reported the pair distribution function for Static-SAXS and SEC-SAXS below as additional information as **Figure S15 c**:

Pair distance distribution function (Static-SAXS) Pair distance distribution function (SEC-SAXS)

Reviewer #2 (Remarks to the Author):

I appreciate the author's effort in conducting the additional experiments; the manuscript looks significantly improved as a result. I still have the following questions:

Why do they double the amount of protein in the dose range study? What is the rationale behind this?

Our concentrations are in the range (100 – 2000 µg/mL). Such interval of concentrations is widely reported in the literature for neutralization/invasion assays against intracellular protozoan parasites using monoclonal antibodies.

1. Weiss GE, et al. The dual action of human antibodies specific to Plasmodium falciparum PfRH5 and PfCyRPA: Blocking invasion and inactivating extracellular merozoites. **PLoS Pathog.** 2023 Sep 15;19(9):e1011182. doi: 10.1371/journal.ppat.1011182.
2. Murugan R, et al. Evolution of protective human antibodies against Plasmodium falciparum circumsporozoite protein repeat motifs. **Nat Med.** 2020 Jul;26(7):1135-1145. doi: 10.1038/s41591-020-0881-9.

I would like a side-by-side comparison between the original TcPOP and the new TcPOP formulation. Does the new formulation elicit a better or comparable response to the vaccine?

The decision to use a higher protein dose compared to previous studies is primarily due to the inclusion of Addavax in the formulation, which aligns with the established protocol used in Prof. Barford's laboratory and has been widely adopted in several peer-reviewed studies. This choice was made to ensure **consistency with existing methodologies** for monoclonal antibody production, rather than as a direct comparison to previous immunization regimes for TcPOP.

We included in the manuscript this information in **lines 184-186**: “ The following vaccine regiment was selected based on previous successful immunisations for the generation of monoclonal antibodies from mouse hybridomas (Knudsen et al., 2021, Knudsen et al., 2022)”.

What is the advantage of this new TcPOP over the previous one? At the very least, could you compare similar parameters to highlight any novelty? Or is this paper primarily focused on protein characterization?

This manuscript reports the first experimentally determined structure of TcPOP, which is **invaluable for future drug design and vaccine development efforts**. For instance, the atomic details presented here offer a starting point for future therapeutic strategies, including the possibility of selectively targeting one conformation over the other for enzyme inhibition.

The primary focus of this study is on the structural characterization of TcPOP, providing essential insights into its dynamicity and the existence of both open and closed conformations. Thanks to the previous suggestions from Reviewer 2, we expanded the parasite studies compared to the first version of the manuscript, opening up new interesting research possibilities, i.e. re-screening from hybridomas for lytic mAbs. However, these studies are beyond the scope of the current manuscript.

Thank you for including the neutralization assay. However, T. cruzi mutants are generally less virulent than the wild type (WT). Could you include the WT for comparison?

The use of fluorescence was consistent with the reviewer's previous request to perform neutralization/invasion assays using an automated readout. However, we recognize his/her concern. We therefore compared the infectivity of WT and reporter strains using Giemsa/Hoechst staining and manual counting.

The image above shows the comparative infectivity of WT and fluorescent parasites, 72 hours post infection with an MOI of 10. There was no significant difference.

This result is in line with multiple *in vitro* and *in vivo* studies that we have carried out over the last decade with parasites where bioluminescent/fluorescent reporters have been integrated into the *T. cruzi* ribosomal locus. Therefore, we opted not to including them as Supplementary data.

As examples:

1. Lewis MD, et al. Bioluminescence imaging of chronic *Trypanosoma cruzi* infections reveals tissue-specific parasite dynamics and heart disease in the absence of locally persistent infection. **Cell Microbiol.** 2014 Sep;16(9):1285-300. doi: 10.1111/cmi.12297.
2. Costa FC, et al. Expanding the toolbox for *Trypanosoma cruzi*: A parasite line incorporating a bioluminescence-fluorescence dual reporter and streamlined CRISPR/Cas9 functionality for rapid *in vivo* localisation and phenotyping. **PLoS Negl Trop Dis.** 2018 Apr 2;12(4):e0006388. doi: 10.1371/journal.pntd.0006388.

3. González S, et al. Short-course combination treatment for experimental chronic Chagas disease. *Sci Transl Med.* **2023** Dec 13;15(726):eadg8105. doi: 10.1126/scitranslmed.adg8105.
4. Jayawardhana S, et al. Benznidazole treatment leads to DNA damage in *Trypanosoma cruzi* and the persistence of rare widely dispersed non-replicative amastigotes in mice. *PLoS Pathog.* **2023** Nov 13;19(11):e1011627. doi: 10.1371/journal.ppat.1011627.
5. Mann GS, et al. Drug-cured experimental *Trypanosoma cruzi* infections confer long-lasting and cross-strain protection. *PLoS Negl Trop Dis.* **2020 Apr** 17;14(4):e0007717. doi:

I also don't see a specific band for the western blot (WB). This is especially important because the monoclonal antibody is expected to recognize something specific in T. cruzi. Here's an example of the expected result with your antigen: DOI: 10.1371/journal.ppat.1012764, Figures S6 and S7.

We appreciate the reviewer's detailed feedback and the reference to Versteeg et al. (doi: 10.1371/journal.ppat.1012764). While we understand the importance of demonstrating specificity using Wb analysis, we would like to clarify several key points regarding our experimental setup and target:

- Our monoclonal antibody (mAb1) **recognises a conformational epitope of TcPOP**, (as shown in Fig 3c), which is not recognised under denaturing conditions. The absence of a band is not a reflection of the antibody's lack of specificity but rather a **limitation of Wb to detect conformational epitopes**. In the paper by Versteeg et al., the Wb analysis uses peptides derived from Tcj2, which are linear sequences that do not rely on maintaining a specific three-dimensional conformation for antibody binding.
- While ELISA could be used to confirm the detection of TcPOP by mAb1 in the parasite lysates (similar to what we did with the recombinant TcPOP), it still would not resolve if the binding is specific to the TcPOP epitope within the parasite. Our employed microscopy techniques, clearly demonstrate that mAb1 is on target (Fig.3 g).
- The exceptional lytic effect of the polyclonal serum provides evidence that the approach presented in this manuscript, in future, can provide a framework for expanding hybridoma screening to identify better mAbs for therapeutic applications.
- Additionally, we would like to remark that exploiting mAb1 for diagnostics was not a specific aim of this work. However, optimising its uses could open a new venue in the diagnostic field for *Trypanosoma* infection / Chagas disease.

Additionally, further studies will be required to identify those mAb(s) responsible of the observed striking neutralisation effects observed in the anti-TcPOP polyclonal serum.

We have integrated all this information in the manuscript in **lines 738-750**:

Western blot experiments suggest that IM1-mAb1 recognises a conformational epitope, while ELISA, BLI and immunofluorescence experiments (IFA) confirmed its specificity for TcPOP, both as recombinant protein or as expressed on whole parasites. IM1-mAb1 therefore has diagnostic potential, although further assessment is beyond the scope of the present study.

Interestingly, the IM1-mAb1 was the weakest binder *in vitro* among the three isolated mAbs, exhibiting faster *kon* and *koff* rates, whilst the other mAbs exhibited much longer *koff* rates. This underscores the importance of testing all mAbs in a cell-invasion scenario that more closely reflects host-pathogen interactions, without excluding any based on *in vitro* studies alone, as interactions with the parasite may

reveal neutralizing potential. Additionally, further studies will be required to identify those mAb(s) responsible of the striking neutralisation effects observed in the anti-TcPOP polyclonal serum (Fig. 1).

Reviewer #3 (Remarks to the Author):

The authors' revision has significantly improved the quality of the manuscript and addressed most of the issues I raised in the previous version. However, I have one remaining concern regarding a minor discrepancy with the global resolution of the open and closed TcPOP structures. As shown in the FSC curve from the PDB validation reports, as well as the global FSC curves in Figure S14, the global resolutions for the open and closed TcPOP structures doesn't appear to be 3.0 Å and 2.8 Å with FSC 0.143 cutoff as stated in the manuscript. I also see that the reported resolution by the authors of open and close TcPOP structures in the PDB validation reports were 3.82 Å and 3.57 Å, respectively. I kindly request that the authors update the manuscript to reflect the resolution values reported in the PDB validation reports.

The previously reported values (2.8 Å and 3.0 Å) were a legacy error from an earlier refinement stage. We have now ensured that all instances in the text, including the abstract and figure legends, reflect the correct values. The revised resolutions from the wwPDB EM Validation Reports are 3.6 Å (closed conformation) and 3.8 Å (open conformation). Additionally, we added in the methods and in Table 1 further information (**lines 398-402**) referencing the usage of DynaMight, which has also been included in Figure S13 (Supplementary Data).

Reviewer #1 (Remarks to the Author):

The authors have addressed the major points raised, and the manuscript has improved considerably compared to the first version. A few considerations for the authors to take into account:

- Please ensure accurate citation of data, particularly for references 3 and 5.
- Additionally, check reference 11.
- Clarify the limitations of using static SAXS, as discussed in the authors' rebuttal letter, and ensure they are included in the current version of the manuscript.

We recognise that the reference 3 and 5 were cited erroneously and that reference 11 was numbered incorrectly. These have now been fixed.

We clarify the limitations of using static SAXS in the main manuscript (lines 257-262):

The discrepancy between the R_g values from Guinier (38 Å) and $P(r)$ analysis (71 Å) in the static SAXS measurements is due to the presence of larger multimers that persist even when extrapolating to zero concentration. As a result, extrapolating to zero concentration cannot fully recover the properties of a species that was never measured in isolation. Given these limitations, we produced the $P(r)$ plots and we relied only on SEC-SAXS data, which directly allowed the isolation of the monomeric species and provides in this case a more reliable approach

Reviewer #2 (Remarks to the Author):

Thank you.

I still want to see the Western blot and the specificity of TcPOP. The assay I'm looking for is similar to what I mentioned in the Versteeg paper. I'd like to load TcPOP protein along with lysates from *T. cruzi* and other parasites like *T. brucei* and *Leishmania*. The Western blot can be performed with and without reducing conditions. Then, I'll use antisera from vaccinated mice to determine if the detection is specific.

Let me know your thoughts!

- Our monoclonal antibody (mAb1) recognises a conformational epitope of TcPOP, (as shown in Fig 3c), which is not recognised under denaturing conditions. The absence of a band is not a reflection of the antibody's lack of specificity but rather a limitation of Wb

to detect conformational epitopes. In the paper by Versteeg et al., the Wb analysis uses peptides derived from Tcj2, which are linear sequences that do not rely on maintaining a specific three-dimensional conformation for antibody binding.

- While ELISA could be used to confirm the detection of TcPOP by mAb1 in the parasite lysates (similar to what we did with the recombinant TcPOP), it still would not resolve if the binding is specific to the TcPOP epitope within the parasite. Our employed microscopy techniques, clearly demonstrate that mAb1 is on target (Fig.3 g).
- The exceptional lytic effect of the polyclonal serum provides evidence that the approach presented in this manuscript, in future, can provide a framework for expanding hybridoma screening to identify better mAbs for therapeutic applications.
- Additionally, we would like to remark that exploiting mAb1 for diagnostics was not a specific aim of this work. However, optimising it uses could open a new venue in the diagnostic field for Trypanosoma infection / Chagas disease.

Based on these considerations, we added in the manuscript (lanes 365-372):

Western blot experiments suggest that IM1-mAb1 recognises a conformational epitope, while ELISA, BLI and immunofluorescence experiments (IFA) confirmed its specificity for TcPOP, both as recombinant purified protein or indirectly when expressed on whole parasites. Although, direct detection was not verified in parasite lysates, IM1-mAb1 still retains diagnostic potential, although further assessment is beyond the scope of the present study. Recognition of the purified TcPOP does not preclude the possibility of cross-reactivity with other T. cruzi proteins, and this will require further confirmation.